# FACTTEST: Factuality Testing in Large Language Models with Finite-Sample and Distribution-Free Guarantees

**Fan Nie** [1]  **Xiaotian Hou** [2]  **Shuhang Lin** [2]  **James Zou** [1]  **Huaxiu Yao** [3]  **Linjun Zhang** [2]

## Abstract

The propensity of large language models (LLMs) to generate hallucinations and non-factual content undermines their reliability in high-stakes domains, where rigorous control over Type I errors (the conditional probability of incorrectly classifying hallucinations as truthful content) is essential. Despite its importance, formal verification of LLM factuality with such guarantees remains largely unexplored. In this paper, we introduce FACTTEST, a novel framework that statistically assesses whether an LLM can provide correct answers to given questions with high-probability correctness guarantees. We formulate hallucination detection as a hypothesis testing problem to enforce an upper bound of Type I errors at user-specified significance levels. Notably, we prove that FACTTEST also ensures strong Type II error control under mild conditions and can be extended to maintain its effectiveness when covariate shifts exist. FACTTEST is distribution-free and and model-agnostic. It works for any number of human-annotated samples and applies to any black-box or white-box LM. Extensive experiments demonstrate that FACTTEST effectively detects hallucinations and enable LLMs to abstain from answering unknown questions, leading to an over 40% accuracy improvement. Code is here.

## 1. Introduction

Large Language Models (LLMs) like ChatGPT (Ouyang et al., 2022; OpenAI, 2024a) have demonstrated substantial advancements across various domains. However, their outputs cannot be fully trusted due to their propensity to generate inaccurate and non-factual information with seemingly high fluency and natural grounding, a challenge known as hallucination (Maynez et al., 2020b; Huang et al., 2023; Ji et al., 2023). This tendency undermines the reliability and trustworthiness of the generated content, highlighting the critical need for robust mechanisms to verify the factuality and correctness of LLM outputs.

Existing approaches to hallucination detection like retrieval-based methods (Thorne et al., 2018b; Gou et al., 2024; Chen et al., 2024) and training-based approaches (Zhang et al., 2023) either rely on external databases or resource-intensive fine-tuning processes, which are often impractical or costly. Therefore, uncertainty estimation has emerged as a zero-resource alternative for hallucination detection (Varshney et al., 2023; Xiong et al., 2024), operating under the premise that hallucinations are intrinsically tied to the model's uncertainty (Huang et al., 2023). However, none of these methods can provide theoretical guarantees for detection, a critical requirement for deploying LLMs in high-stakes domains (Kumar et al., 2023) where precise control of Type I errors (incorrectly flagging a hallucination as truthful content) is needed for decision-making. For instance, incorrect medical diagnoses or the provision of uncertain legal advice could result in detrimental consequences.

To address these limitations, we introduce FACTTEST, a framework that statistically evaluates whether an LLM can reliably generate correct answers to given questions with provable correctness guarantees. We formulate the factuality testing within a hypothesis testing framework to theoretically control the Type I error while minimizing the Type II error. Leveraging the fundamental connection between Neyman-Pearson (NP) classification and statistical testing (Tong et al., 2018; Tong, 2013; Scott & Nowak, 2005), we define a score function to quantify correctness and determine an appropriate threshold based on a calibration dataset. This allow LLMs to refuse unknown questions and control the false positive rate for any score function.

Our theoretical analysis shows that FACTTEST achieves strong power control when the score function effectively captures output correctness, ensuring not only Type I error control but also a low Type II error. We also establish a formal connection between NP classification and PAC-style conformal prediction (Vovk, 2012) within our framework,

[1]Stanford University, USA [2]Rutgers University, USA [3]The University of North Carolina at Chapel Hill, USA. Correspondence to: Linjun Zhang <linjun.zhang@rutgers.edu>.

*Proceedings of the 42nd International Conference on Machine Learning*, Vancouver, Canada. PMLR 267, 2025. Copyright 2025 by the author(s).

unifying two seemingly distinct methodologies and expanding the applicability of conformal prediction. On the other hand, recognizing that the i.i.d. assumption underlying statistical tests may not always hold in practice, we enhance the robustness of our framework by incorporating an extension to handle covariate shifts through the estimation of density ratios and the use of rejection sampling. Our approach is model-agnostic and does not rely on specific data distribution assumptions, making it broadly applicable to any language model. Importantly, it is easy to use and works for any number of human-annotated samples.

To the best of our knowledge, this study is the first to introduce statistical factuality testing for large language models. Evaluation of our proposed framework on question-answering (QA) benchmarks demonstrate several key advantages of our approach: (1) it consistently outperforms base models by a substantial margin without requiring additional training or external data sources; (2) it surpasses fine-tuned baselines by a large margin while utilizing only half of the training data; and (3) it maintains superior performance on out-of-distribution testing data. Notably, the theoretical guarantees of our method remain valid even when the i.i.d. assumption is violated. **We summarize the main contributions below**.

- We propose FACTTEST, a novel statistical testing framework that evaluates the factuality of LLMs while enabling them to decline unknown questions with user-specified Type I error guarantees.

- We establish the first formal connection between NP classification and PAC-style conformal prediction within factuality testing, providing a unified framework with rigorous guarantees.

- We prove that FACTTEST achieves strong power control under mild conditions, ensuring a low Type II error. This power analysis is broadly applicable to standard NP classification problems, not limited to this setting.

- We extend FACTTEST to accommodate covariate shifts by approximating density ratios and employing rejection sampling, enhancing its robustness for real-world applications.

- We demonstrate that FACTTEST effectively detects hallucinations while maintaining Type I error below user-specified significance levels, achieving an over 40% improvement in accuracy compared to base models without any fine-tuning. Additionally, it surpasses training-based baselines by 30% using only half of the fine-tuning data.

## 2. Statistical Factuality Testing

In this section, we formulate the evaluation of factuality in LLMs as a hypothesis testing problem and introduce our FACTTEST framework to overcome hallucination issues.

### 2.1. Problem Formulation

We consider a text generation task in which a language model $M$ will generate its answers $M(q)$ based on a question $q$. Our goal is to statistically evaluate whether $M$ can correctly answer $q$. We formulate this objective as a hypothesis testing problem with the following hypotheses:

$H_0$ : The model $M$ does not answer the question $q$ correctly.

$H_1$ : The model $M$ answers the question $q$ correctly.

For any question-answer pair $(q, a)$ with $a$ to be one of the correct answer for question $q$, we apply $M$ to generate an answer $M(q)$. The question-generated answer pair $(q, M(q))$ is deemed correct if the null hypothesis $H_0$ is rejected, i.e., $M(q)$ aligns with $a$; otherwise, it is deemed incorrect. Let $P_0$ represent the distribution of $(q, M(q))$ given $M(q)$ differs from the correct answer, and let $P_1$ denote the distribution of $(q, M(q))$ when $M(q)$ aligns with the correct answer. Given a dataset $\{(q_1, a_1), ..., (q_n, a_n)\} \subset \mathcal{Q} \times \mathcal{A} \overset{\text{i.i.d.}}{\sim} P_{q,a}$ comprising $n$ question-answer pairs with $\mathcal{Q}, \mathcal{A}$ to be the set of all possible questions and answers, respectively, and $P_{q,a}$ is a distribution of $\mathcal{Q} \times \mathcal{A}$, we apply $M$ to generate answers for all the $n$ questions, resulting in the set $\mathcal{D} = \{(q_1, M(q_1), a_1), \ldots, (q_n, M(q_n), a_n)\}$. Since the distribution $P_{M(q)|q}$ of $M(q)$ produced by $M$ given the question $q$ is fully determined by $M$ and independent of $a$, we know $\mathcal{D} \overset{\text{i.i.d.}}{\sim} P_{q,M(q),a} = P_{q,a} P_{M(q)|q}$. Then our goal is to construct a predictor $\hat{f}_\alpha : \mathcal{Q} \times \mathcal{A} \to \{0, 1\}$ that classifies a pair $(q, M(q))$ as correct (output 1) or incorrect (output 0) while ensuring that the false positive rate, or Type I error, does not exceed a pre-specified significance level $\alpha$. Formally, we seek $\hat{f}_\alpha$ such that the error of predicting incorrect $(q, M(q))$ as correct is below level $\alpha$ with probability at least $1 - \delta$, i.e.,

$$\mathbb{P}_{\mathcal{D}}(\mathbb{P}_{(q,M(q)) \sim P_0}(\hat{f}_\alpha(q, M(q)) = 1) > \alpha) \le \delta. \quad (1)$$

where $\delta$ denotes the allowable probability of exceeding $\alpha$. Note that given any question $q$, the answer $M(q)$ generated by $M$ is randomized. While the distribution of $M(q)$ is fully determined by $q$, the realization $M(q)$ involves additional sampling randomness independent of $q$. Taking $(q, M(q))$ as input to $\hat{f}_\alpha$, we enable the predictor to utilize information from the question $q$, the distribution of $M(q)$ (by asking $M$ the same question $q$ multiple times), and the current realization $M(q)$ of the produced answer.

### 2.2. Finite-sample and Distribution-free Type I Error Control

**Calibration Dataset Construction.** Following the methodology of Zhang et al. (2023), we adopt a supervised identification strategy to partition the dataset $\mathcal{D}$ into a correct subset $\mathcal{D}_1$ and an incorrect subset $\mathcal{D}_0$.

Specifically, for each question-generated answer pair

$(q_i, M(q_i))$ in $\mathcal{D}$, we define an indicator variable $y_i \in \{0,1\}$ to indicate the correctness of $M(q_i)$ such that

$$y_i = \begin{cases} 1, & \text{if } M(q_i) \text{ aligns with the true answer } a_i, \\ 0, & \text{otherwise.} \end{cases}$$

Based on these indicators, the dataset is divided into:

$$\mathcal{D}_1 = \{(q_i, M(q_i)) \in \mathcal{Q} \times \mathcal{A} : y_i = 1, i \in [n]\},$$
$$\mathcal{D}_0 = \{(q_i, M(q_i)) \in \mathcal{Q} \times \mathcal{A} : y_i = 0, i \in [n]\}.$$

Note that the construction of the indicator variable $y = \mathbb{I}(M(q)$ aligns with $a)$ for $(q, M(q), a) \sim P_{q,M(q),a}$ defines a distribution $P_{q,M(q),a,y}$, then the data $\{(q_i, M(q_i), y_i)\}_{i=1}^n$ are i.i.d. samples from $P_{q,M(q),y}$ over all possible combinations of $(q, M(q), y)$, and the distributions of $\mathcal{D}_0$ and $\mathcal{D}_1$ are $P_0 = P_{q,M(q)|y=0}$ and $P_1 = P_{q,M(q)|y=1}$, respectively.

**Correctness Predictor based on Score Function.** Suppose there is a score function $\hat{\eta} : \mathcal{Q} \times \mathcal{A} \to \mathbb{R}$ that measures the correctness of $(q, M(q))$. The value is expected to be large if $M$ has the ability to provide a factual answer. The predictor $\hat{f}_\alpha(q, M(q))$ can then be defined as:

$$\hat{f}_\alpha(q, M(q)) = \mathbb{I}(\hat{\eta}(q, M(q)) > \hat{\tau}_\alpha) \tag{2}$$

where $\mathbb{I}$ is the indicator function and $\hat{\tau}_\alpha$ is a threshold to be determined. The task thus reduces to selecting a threshold $\hat{\tau}_\alpha$ that satisfies the requirement in Eq. 1:

$$\mathbb{P}_\mathcal{D}(\mathbb{P}_{(q,M(q))\sim P_0}(\hat{\eta}(q, M(q)) > \hat{\tau}_\alpha) > \alpha) \leq \delta. \tag{3}$$

**Calibration and Threshold Selection** To determine the appropriate threshold $\hat{\tau}_\alpha$, we utilize the calibration subset $\mathcal{D}_0$. Denote the $n_0$ samples in $\mathcal{D}_0$ by $\mathcal{D}_0 = \{(q_i^{(0)}, M(q_i^{(0)})) : i \in [n_0]\}$. For each calibration sample $(q_i^{(0)}, M(q_i^{(0)})) \in \mathcal{D}_0$, we compute the score $T_i = \hat{\eta}(q_i^{(0)}, M(q_i^{(0)}))$. We then order these scores in ascending order to obtain the order statistics $T_{(1)} \leq \ldots \leq T_{(n_0)}$, and set $T_{(n_0+1)} = +\infty$. Motivated by the seminal works (Vovk, 2012) on the PAC-style conformal prediction and (Tong et al., 2018) on Neyman-Pearson classification, if we set the threshold $\hat{\tau}_\alpha$ to be the $k$th smallest score $T_{(k)}$, the probability for $\hat{f}_\alpha$ to have type I error greater than $\alpha$ can be controlled in a distribution-free and finite-sample manner,

$$\mathbb{P}_\mathcal{D}(\mathbb{P}_{(q,M(q))\sim P_0}(\hat{\eta}(q, M(q)) > T_{(k)}) > \alpha)$$
$$\leq \sum_{j=k}^{n_0} \binom{n_0}{j} (1-\alpha)^j \alpha^{n_0-j} \triangleq v(k), \quad k \in [n_0+1], \tag{4}$$

when $k = n_0+1$, $v(k)$ is defined to be 0. We then determine $\hat{k}$ as

$$\hat{k} = \min\{k \in [n_0+1] : v(k) \leq \delta\}, \tag{5}$$

The threshold is then set to: $\hat{\tau}_\alpha = T_{(\hat{k})}$. Note that $\hat{\tau}_\alpha$ is well defined for any $n_0$, ensuring Type I error control irrespective of the calibration sample size $n$. Specifically, when $n_0$ is small such that $v(n_0) > \delta$, the threshold becomes $\hat{\tau}_\alpha = T_{(n_0+1)} = +\infty$, causing $\hat{f}_\alpha$ to conservatively classify all pairs $(q, M(q))$ as incorrect, thereby abstaining from answering any question. The derivation is deferred to Appendix. A.

**Theorem 2.1.** *For any $n \in \mathbb{N}_+$, with probability at least $1 - \delta$, the constructed classifier $\hat{f}_\alpha$ has type I error below $\alpha$, i.e.,*

$$\mathbb{P}_\mathcal{D}(\mathbb{P}_{(q,M(q))\sim P_0}(\hat{f}_\alpha(q, M(q)) = 1) \leq \alpha) \geq 1 - \delta.$$

With the determined threshold $\hat{\tau}_\alpha$, the predictor $\hat{f}_\alpha(q, M(q)) = \mathbb{I}(\hat{\eta}(q, M(q)) > \hat{\tau}_\alpha)$ is formally defined. This classifier ensures that, for a given significance level $\alpha$, the Type I error is controlled below $\alpha$ with high probability $1 - \delta$. Consequently, when $\hat{\eta}(q, M(q)) \geq \hat{\tau}_\alpha$, we reject the null hypothesis $H_0$ and assert that the model $M$ can answer the question $q$ correctly. Otherwise, the model will output an acknowledgment of uncertainty.

### 2.3. Type II Error Control

The effectiveness of FACTTEST not only hinges on Type I error control but also on ensuring sufficient statistical power to detect true positives. We then analyze the Type II error of the constructed classifier, which is the probability of misclassifying correct $(q, M(q))$ from $P_1$ as incorrect.

Denote $\eta(q, M(q)) = \mathbb{P}_{y\sim P_{y|q,M(q)}}(y = 1|q, M(q))$ to be the conditional probability that $M(q)$ aligns with the correct answer $a$ given any question $q$ and the generated answer $M(q)$. Note that a question $q$ may have multiple correct answers and $a$ is just one realization from $P_{a|q}$. Therefore, $a$, and thus $y$, may still be random given $(q, M(q))$, implying $\eta(q, M(q))$ may take value in $(0, 1)$. For any classifier $f$, we set $\mathcal{R}_0(f) = \mathbb{P}_{(q,M(q))\sim P_0}(f(q, M(q)) = 1)$ (resp. $\mathcal{R}_1(f) = \mathbb{P}_{(q,M(q))\sim P_1}(f(q, M(q)) = 0)$) to be the Type I error (resp. Type II error). It follows from Theorem 1 in (Tong, 2013) that the Bayes optimal classifier $f_\alpha^*$

$$f_\alpha^* \in \underset{f:\mathcal{Q}\times\mathcal{A}\to\{0,1\}}{\arg\min} \mathcal{R}_1(f) \quad \text{s.t.} \quad \mathcal{R}_0(f) \leq \alpha$$

has the form $f_\alpha^*(q, M(q)) = \mathbb{I}(\eta(q, M(q)) > \tau_\alpha)$ for some $\tau_\alpha \in [0, 1]$. Therefore, $f_\alpha^*$ is the optimal rule for detecting incorrect answers and $\eta$ is an optimal choice of the score function.

Suppose there exist an increasing function $H$ and $\epsilon_\eta > 0$ such that $\|H \circ \hat{\eta} - \eta\|_\infty \leq \epsilon_\eta$, where $H \circ \hat{\eta}(q, M(q)) = H(\hat{\eta}(q, M(q)))$ is the composition of $H$ and $\hat{\eta}$. Let $p_y = \mathbb{P}_{y\sim P_y}(y = 1)$ denote the marginal probability that $M$ is correct. We define:

$$\xi_\alpha = \frac{\tau_\alpha(1-p_y)}{(1-\tau_\alpha)p_y}, \quad \alpha' = \alpha - c\sqrt{\frac{\alpha}{n_0}\log\frac{1}{\delta}},$$

$$\epsilon_\tau = \tau_{\alpha'} - \tau_\alpha + \epsilon_\eta,$$

for some constant $c > 0$. If we denote $G_\alpha(\epsilon) = \mathbb{P}_{(q,M(q))\sim P_0}(|\eta(q, M(q)) - \tau_\alpha| \le \epsilon)$ to be the probability measure around the classification boundary of $f_\alpha^*$, then the following theorem states that as long as the score function $\hat{\eta}$ measures the level of correctness of $M$, the type II error of our algorithm is small.

**Theorem 2.2.** *If $\hat{\eta}(q, M(q))$ is a continuous random variable with $(q, M(q)) \sim P_0$, $\alpha \gtrsim \frac{\log 1/\delta}{n_0}$ and $\tau_\alpha + \epsilon_\tau + \epsilon_\eta < 1$, then with probability at least $1 - 2\delta$, we have*

$$\mathcal{R}_1(\hat{f}_\alpha) - \mathcal{R}_1(f_\alpha^*)$$

$$\lesssim \xi_\alpha\sqrt{\frac{\alpha}{n_0}\log\frac{1}{\delta}} + \frac{(1-p_y)(\epsilon_\tau+\epsilon_\eta)}{p_y(1-\tau_\alpha-\epsilon_\tau-\epsilon_\eta)^2}G_\alpha(\epsilon_\tau+\epsilon_\eta).$$

## 3. Extension of FACTTEST to Covariate Shifts

The threshold selection procedure developed in Section 2 relies on the assumption that the calibration dataset $\mathcal{D}_0 = \{(q_i, M(q_i)) \in \mathcal{Q} \mid y_i = 0\}$ follows the target distribution $P_0$ of incorrect question-generated answer pairs. However, labeled samples from the target distribution may not always be available in practice. Instead, people may use the labeled data that they believe to be similar to the target distribution, which necessitates methods to handle distribution shifts. In this section, we study the case of covariate shift, where the distribution of the question-generated answer pairs in the calibration data differs from that in the target distribution, while the conditional distribution of $y$ given $(q, M(q))$ remains the same.

### 3.1. Setup

Suppose we observe $n$ samples $\mathcal{D} = \{(q_i, M(q_i), y_i) : i \in [n]\}$ from the source distribution $\tilde{P}_{q,M(q),y}$. We assume $P_{y|q,M(q)} = \tilde{P}_{y|q,M(q)}$ but $P_{q,M(q)} \ne \tilde{P}_{q,M(q)}$, i.e., the distribution of questions changes but the oracle rule of detecting incorrect answers remains. Following Section 2, we split $\mathcal{D}$ into a correct subset $\mathcal{D}_1 = \{(q_i, M(q_i)) : y_i = 1, i \in [n]\} = \{(q_i^{(1)}, M(q_i^{(1)})) : i \in [n_1]\}$ and an incorrect subset $\mathcal{D}_0 = \{(q_i, M(q_i)) : y_i = 0, i \in [n]\} = \{(q_i^{(0)}, M(q_i^{(0)})) : i \in [n_0]\}$. We denote the distribution of $\mathcal{D}_0, \mathcal{D}_1$ to be $P_0, \tilde{P}_1$, respectively. We further denote the density ratio between the target distribution $P_0$ of incorrect question-generated answer pair and the source distribution $\tilde{P}_0$ to be $w(q, M(q)) = \frac{dP_0}{d\tilde{P}_0}(q, M(q))$. In this section, we assume $w$ is known and satisfies $w(q, M(q)) \le B$ for all $(q, M(q)) \in \mathcal{Q} \times \mathcal{A}$.

### 3.2. Type I Error Control under Covariate Shift

To extend the procedure in Section 2 to the covariate shift setting, we take an additional step to transform the samples in $\mathcal{D}_0$ from $\tilde{P}_0$ to $P_0$ distributed random variables by rejection sampling.

In the first step, we generate $n_0$ uniform random variables $U_1, \ldots, U_{n_0} \overset{\text{i.i.d.}}{\sim} \text{Unif}[0, B]$ and select the indexes $\mathcal{I} = \{i \in [n_0] : U_i \le w(q_i^{(0)}, M(q_i^{(0)}))\}$. If we collect all the samples in $\mathcal{D}_0$ with indexes in $\mathcal{I}$ to form $\tilde{\mathcal{D}}_0 = \{(q_i^{(0)}, M(q_i^{(0)})) : i \in \mathcal{I}\} \overset{\triangle}{=} \{(\tilde{q}_i, M(\tilde{q}_i)) : i \in [\tilde{n}_0]\}$. Then it will be shown in Appendix A that given the selection $\mathcal{I}$ by rejection sampling, the selected samples $\tilde{\mathcal{D}}_0$ follow the target distribution $P_0$ i.e., $\tilde{\mathcal{D}}_0 \mid \mathcal{I} \overset{\text{i.i.d.}}{\sim} P_0$.

In the second step, we apply the procedure introduced in Section 2 to the incorrect subset $\tilde{\mathcal{D}}_0$. Specifically, given the incorrect subset $\tilde{\mathcal{D}}_0$, we calculate the scores $\tilde{T}_i = \hat{\eta}(\tilde{q}_i, M(\tilde{q}_i))$ and order them in increasing order to get $\tilde{T}_{(1)} \le \ldots \le \tilde{T}_{(\tilde{n}_0)}$, and set $\tilde{T}_{(\tilde{n}_0+1)} = +\infty$. Then we set the threshold $\hat{\tau}_\alpha$ to be $\tilde{T}_{(\hat{k})}$, with $\hat{k}$ satisfies

$$\hat{k} = \min\{k \in [\tilde{n}_0+1] : \tilde{v}(k) \le \delta\},$$

$$\tilde{v}(k) = \sum_{j=k}^{\tilde{n}_0}\binom{\tilde{n}_0}{j}(1-\alpha)^j\alpha^{\tilde{n}_0-j}, \quad \tilde{v}(\tilde{n}_0+1) = 0.$$

Since $\tilde{\mathcal{D}}_0 \mid \mathcal{I} \overset{\text{i.i.d.}}{\sim} P_0$, theoretical results in Section 2 can be directly applied here. Due to limited space, we control the Type I error as follows.

**Theorem 3.1.** *With probability at least $1-\delta$, the constructed classifier $\hat{f}_\alpha(q, M(q)) = \mathbb{I}(\hat{\eta}(q, M(q)) > \tilde{T}_{(\hat{k})})$ has type I error below $\alpha$, i.e.*

$$\mathbb{P}_\mathcal{D}(\mathbb{P}_{(q,M(q))\sim P_0}(\hat{f}_\alpha(q, M(q)) = 1) \le \alpha) \ge 1 - \delta.$$

## 4. Connection with Conformal Prediction

In Sec. 2, we formulate our method within the NP classification framework by defining an NP classifier for error control. In this section, we explore the inherent relationship between NP classification (Tong et al., 2018) and PAC-style conformal prediction (Vovk, 2012), demonstrating that FACTTEST can also be formulated within the lens of conformal prediction. To the best of our knowledge, this work presents *the first formal articulation of the connection between NP classification and PAC-style conformal prediction.*

**Challenges of conventional conformal prediction.** Recent works applying conformal prediction to LLMs (Kumar et al., 2023; Quach et al., 2024; Li et al., 2024) have primarily focused on answer set coverage guarantee. A naive application of prediction sets to hallucination detection might involve verifying whether outputs fall within the prediction set. However, this approach is fundamentally flawed for

the hallucination control problem formulated in Section 2.1, because enforcing $(1-\alpha)$ coverage guarantee inherently tolerates incorrect answers within the prediction set, leading to large Type I errors. In an extreme scenario, ensuring 100% coverage would require including all possible answers, resulting in a 100% Type I error rate. More empirical results can be seen in App. D.1.

**PAC-style conformal prediction.** Suppose we are given a dataset $\mathcal{D}^{\mathrm{cp}} = \{X_i : i \in [n]\} \overset{\text{i.i.d.}}{\sim} P_X^{\mathrm{cp}}$, where each sample $X_i \in \mathcal{X}$ follows the distribution $P_X^{\mathrm{cp}}$. PAC-style conformal prediction (Vovk, 2012) aims to construct a predictive set $\Gamma$ for an independent random element $X \sim P_X^{\mathrm{cp}}$ such that

$$\mathbb{P}_{\mathcal{D}^{\mathrm{cp}}}\big(\mathbb{P}_{X \sim P_X^{\mathrm{cp}}}(X \notin \Gamma) > \alpha\big) \le \delta. \tag{6}$$

To this end, (Vovk, 2012) proposes to use a conformity score $A : \mathcal{X} \to \mathbb{R}$ for measuring how well $X$ conforms to the dataset $\mathcal{D}^{\mathrm{cp}}$. Given any conformity score $A$, we define the conformity scores of samples $X_i$ in $\mathcal{D}^{\mathrm{cp}}$ as $S_i = A(X_i)$. Then a $p$-value $p(X)$ is defined by

$$p(X) = \frac{|\{S_i : S_i \le S(X)\}| + 1}{n+1}.$$

Finally, for some $\epsilon > 0$, the predictive set $\Gamma$ is defined by

$$\Gamma^\epsilon(\mathcal{D}^{\mathrm{cp}}) = \{x \in \mathcal{X} : p(x) > \epsilon\}. \tag{7}$$

It follows from (Vovk, 2012) that as long as

$$\sum_{j=0}^{\lfloor \epsilon(n+1)-1 \rfloor} \binom{n}{j} \alpha^j (1-\alpha)^{n-j} \le \delta,$$

the coverage guarantee 6 is satisfied.

**Duality of PAC-conformal prediction and FACTTEST.** If we set the conformity score $A$ to be $-\hat{\eta}$, where $\hat{\eta}$ is introduced in Sec. 2.2 and construct the predictive set $\Gamma^\epsilon$ defined in 7 using data $\mathcal{D}^{\mathrm{cp}} = \mathcal{D}_0$, then the predictive set $\Gamma^\epsilon(\mathcal{D}_0)$ for $(q, M(q))$ and the classifier $\hat{f}$ in Sec. 2.2 are closely related as stated in the following lemma:

*Lemma* 1. If we set $\epsilon$ such that $\lfloor \epsilon(n_0+1)-1 \rfloor = n_0 - \hat{k}$, then

$$\hat{f}_\alpha(q, M(q)) = \mathbb{I}\big((q, M(q)) \notin \Gamma^\epsilon(\mathcal{D}_0)\big).$$

The proof of this lemma is straightforward and thus omitted. This equivalence highlights that *NP classification-based threshold selection aligns with the membership determination in PAC-style conformal prediction*, which is akin to the well-known duality between confidence interval and hypothesis testing. Under our setting, conformal prediction aims to construct a confidence set for $(q, M(q))|y = 0$, while Neyman-Pearson classification aims to test whether $y = 0$ or not given $(q, M(q))$.

**Distinctive Features of FACTTEST.** To the best of our knowledge, we are the first to introduce statistical factuality testing for LLMs. In addition, while FACTTEST can be interpreted through both NP classification and PAC-style conformal prediction, our key distinction lies in our detailed analysis of Type II error, enabling the control of both Type I and II errors. Further, we identify the optimal score function for constructing the best classifier with minimal Type II error, a topic that has not been explored in the PAC conformal prediction literature.

# 5. Experiments

In this section, we empirically investigate FACTTEST in addressing the hallucination problem of LLMs, focusing on the following questions: **Q1:** Can FACTTEST improve the accuracy and lead to more factual LLMs? **Q2:** Can FACTTEST effectively control the Type I error? **Q3:** Can FACTTEST generalize well when covariate shifts exist?

## 5.1. Experimental Setups

**Datasets.** Following R-Tuning (Zhang et al., 2023), we conduct experiments on knowledge-extensive QA tasks, categorized into two generation tasks. More details are provided in Appendix C.2.

- *Question-Answering:* Given a question, the model directly predicts its answer. We include **ParaRel** (Elazar et al., 2021) and **HotpotQA** (Yang et al., 2018). For experiments considering distirbution shifts, we utilize **ParaRel-OOD** as the testing dataset, which comprises questions from different domains compared with ParaRel.

- *Multiple-Choice:* Given a question with several choices, the model chooses one option among A, B and C. We include **WiCE** (Kamoi et al., 2023) and **FEVER** (Thorne et al., 2018a).

Evaluating whether $M(q)$ aligns with the answer $a$ depends on the datasets. For question-answering datasets, we verify whether the first few tokens contain $a$. For multiple-choice datasets, we check whether $M(q)$ matches exactly $a$.

**Score Functions.** We can fit a model to predict correctness or use any off-the-shelf function to serve as $\hat{\eta}$. Particularly, we introduce three entropy-based certainty functions. Details about the score functions are deferred to Appendix C.3.

- **Vanilla Entropy (VE)**: We query the model $M$ $k$ times and calculate the entropy across $k$ answers.

$$VE(q, M(q)) = -\sum_{j=1}^{k} p(M(q)_j|q) \log p(M(q)_j|q),$$

$$\hat{\eta}(q, M(q)) = -VE(q, M(q)). \tag{8}$$

where $p(M(q)_j|q)$ is the frequency of a predicted answer

Table 1: The accuracy performance (%) of FACTTEST compared to baselines on question-answering and multiple-choice datasets using a significance level of $\alpha = 0.05$. For brevity, FACTTEST is abbreviated as FTEST. The notation FTEST-ve$_{15}$ denotes the use of a vanilla entropy score function with 15 generated outputs.

| Dataset | Model | Base | SelfCheckGPT | FTEST-ve$_5$ | FTEST-ve$_{10}$ | FTEST-ve$_{15}$ | FTEST-se$_5$ | FTEST-se$_{10}$ | FTEST-se$_{15}$ | FTEST-kle$_{15}$ |
|---|---|---|---|---|---|---|---|---|---|---|
| ParaRel | OpenLLaMA-3B | 36.66 | 53.60 | 60.54 | 66.75 | 67.28 | 60.10 | 62.50 | 67.26 | **78.45** |
| | OpenLLaMA-7B | 40.38 | 60.05 | 74.92 | 79.87 | **80.29** | 65.53 | 71.40 | 65.23 | 76.83 |
| | OpenLLaMA-13B | 42.21 | 59.62 | 77.37 | 77.31 | 79.41 | 73.49 | 68.89 | 73.09 | **83.84** |
| HotpotQA | OpenLLaMA-3B | 25.72 | 36.42 | 50.81 | 55.19 | 53.75 | 45.37 | 52.55 | 52.66 | **55.35** |
| | OpenLLaMA-7B | 28.63 | 39.16 | 56.06 | 59.69 | **60.67** | 51.48 | 53.75 | 56.56 | 60.66 |
| | LLaMA-13B | 30.83 | 41.78 | 51.49 | 54.41 | 49.74 | 55.41 | 57.18 | **60.69** | 54.49 |
| WiCE | OpenLLaMA-3B | 64.72 | 66.36 | 67.65 | 75.00 | 68.18 | 64.71 | **85.71** | 66.67 | – |
| | OpenLLaMA-7B | 72.96 | 75.00 | 50.00 | 55.88 | 47.37 | 90.00 | **100.0** | 90.00 | – |
| | LLaMA-13B | 56.89 | 57.39 | 63.33 | 45.45 | 44.44 | **100.0** | 82.35 | 90.00 | – |
| FEVER | OpenLLaMA-3B | 39.74 | 41.97 | 60.24 | 62.50 | 41.72 | 82.40 | 79.23 | **83.90** | – |
| | LLaMA-7B | 35.99 | 40.89 | 43.92 | 50.94 | **51.38** | 28.69 | 33.12 | 33.27 | – |
| | LLaMA-13B | 32.15 | 41.25 | 38.74 | 42.48 | 46.07 | 49.92 | **54.17** | 52.23 | – |

Table 2: The Type I error of FACTTEST on question-answering and multiple-choice datasets when $\alpha = 0.05$.

| Dataset | Model | FTEST-ve$_5$ | FTEST-ve$_{10}$ | FTEST-ve$_{15}$ | FTEST-se$_5$ | FTEST-se$_{10}$ | FTEST-se$_{15}$ | FTEST-kle$_{15}$ |
|---|---|---|---|---|---|---|---|---|
| ParaRel | OpenLLaMA-3B | 0.0508 | 0.0467 | 0.0513 | 0.0479 | 0.0520 | 0.0486 | 0.0342 |
| | OpenLLaMA-7B | 0.0225 | 0.0093 | 0.0145 | 0.0393 | 0.0394 | 0.0435 | 0.0400 |
| | OpenLLaMA-13B | 0.0192 | 0.0087 | 0.0302 | 0.0341 | 0.0477 | 0.0337 | 0.0331 |
| HotpotQA | OpenLLaMA-3B | 0.0242 | 0.0247 | 0.0272 | 0.0289 | 0.0319 | 0.0297 | 0.0309 |
| | OpenLLaMA-7B | 0.0273 | 0.0298 | 0.0295 | 0.0344 | 0.0298 | 0.0308 | 0.0266 |
| | LLaMA-13B | 0.0200 | 0.0226 | 0.0367 | 0.0278 | 0.0300 | 0.0286 | 0.0353 |
| WiCE | OpenLLaMA-3B | 0.0325 | 0.0089 | 0.0207 | 0.0175 | 0.0029 | 0.0118 | – |
| | OpenLLaMA-7B | 0.0694 | 0.0579 | 0.0617 | 0.0077 | 0.0 | 0.0039 | – |
| | LLaMA-13B | 0.0266 | 0.0290 | 0.0363 | 0.0 | 0.0072 | 0.0024 | – |
| FEVER | OpenLLaMA-3B | 0.0164 | 0.0005 | 0.0217 | 0.0570 | 0.0471 | 0.0496 | – |
| | LLaMA-7B | 0.0598 | 0.0081 | 0.0329 | 0.0392 | 0.0495 | 0.0495 | – |
| | LLaMA-13B | 0.0172 | 0.0383 | 0.0293 | 0.0459 | 0.0518 | 0.0552 | – |

$M(q)_j$ given a question $q$.

- **Semantic Entropy (SE)**: Kuhn et al. (2023) measures uncertainty in natural language generation by accounting for the probability distribution over distinct meanings rather than individual token sequences.

- **Kernel Language Entropy (KLE)**: Nikitin et al. (2024) quantifies uncertainty by using semantic similarity kernels between generated answers, allowing for a more nuanced estimation of uncertainty. Notably, this function does not apply to multiple-choice datasets and we only employ it on ParaRel and HotpotQA.

**Models.** In main experiments, we focus on distribution-free settings, where models do not make specific assumptions about the underlying distribution. We include *Base* and *Self-checkgpt* (Manakul et al., 2023) as baselines. *Base* evaluates the original model on the entire test set without any modifications, while *Selfcheckgpt* and FACTTEST are assessed only on questions for which they can confidently provide answers. We utilize three score functions to implement 9 variants of FACTTEST. Specifically, FACTTEST-ve$_k$, FACTTEST-se$_k$ and FACTTEST-kle$_k$ correspond to using VE, SE and KLE as score functions, respectively, where $k$ denotes the number of sampled outputs for a given question.

To facilitate comparison with training-based methods, we randomly split our training dataset, allocating half for instruction-tuning and the remaining half to construct the calibration dataset. We use 15-generation SE as the score function, referring to this variant as FACTTEST-t. For comparative analysis, we include *R-Tuning* (Zhang et al., 2023) as our primary baseline, evaluating it on the subset of questions that it is willing to answer. We also consider *Finetune-All* and *Finetune-Half*, which undergo instruction-tuning using the entire and half of the original training dataset, respectively, and are evaluated on the entire test set.

To evaluate the applicability of our framework on black-box APIs, we further implement FACTTEST on GPT-4o Mini, GPT-4o (OpenAI, 2024b), Gemini-1.5 and Claude-3.5 (Anthropic, 2024).

**Metrics.** For models that could only output either the answer or an unknown expression, we calculate accuracy on questions that the model is willing to answer:

$$\text{Acc} = \frac{\text{\# of correctly and willingly answered questions}}{\text{\# of willingly answered questions}}. \quad (9)$$

Besides, we also include Type I error (False Positive Rate, FPR), and Type II error (False Negative Rate, FNR), as our

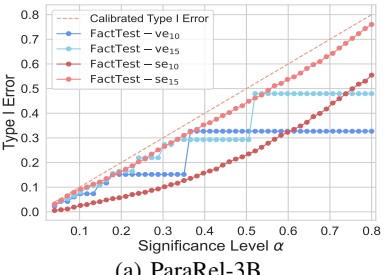 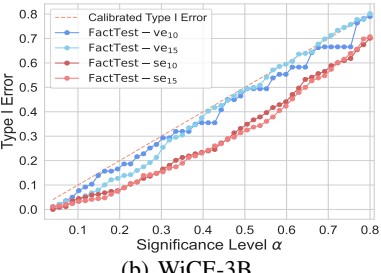 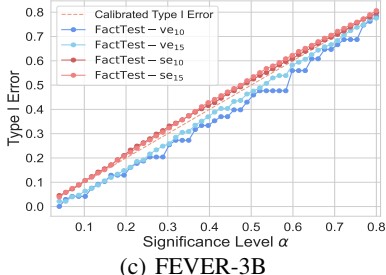

(a) ParaRel-3B       (b) WiCE-3B       (c) FEVER-3B

Figure 1: FACTTEST can control the Type I error given a significance level $\alpha$. The caption of each sub-figure consists of the dataset name and the model size.

evaluation metrics.

**Implementation.** We choose LLaMA-7B, LLaMA-13B (Touvron et al., 2023), OpenLLaMA-3B, OpenLLaMA-7B, OpenLLaMA-13B (Geng & Liu, 2023) as base models in our main text. Experiments involving Mistral-7B (Jiang et al., 2023), LLaMA-3.2-3B-Instruct (Dubey et al., 2024) and Tulu2-7B (Ivison et al., 2023) are deferred to App. D.6. The temperature is set to 0 for evaluation and 0.7 for calculating score functions. We follow Zhang et al. (2023) to use LMFlow (Diao et al., 2023) to conduct instruction tuning, setting epoch to 1 and learning rate to $2e^{-5}$. All experiments are implemented on 4 Nvidia H100-80GB GPUs.

## 5.2. Main Experimental Results

We first conduct in-distribution experiments on QA datasets ParaRel, HotpotQA, WiCE and FEVER.

**Main Performance.** The accuracy results are presented in Table 1, where the significance level $\alpha$ for FACTTEST is set to 0.05. Additional experimental results for other significance levels (e.g., $\alpha = 0.10$) are provided in Appendix D.2. Analysis of the results reveals that FACTTEST significantly outperforms baselines by a substantial margin in terms of accuracy on the questions it is willing to answer. Notably, FACTTEST can yield an over 40% accuracy improvement on WiCE and FEVER. These results demonstrate that FACTTEST is more reliable when making predictions and is capable of refusing unknown answers.

**Type I Error.** Table 2 represents the Type I error, or FPR, of FACTTEST when $\alpha$ is set to 0.05. Figure 1 depicts the FPR-$\alpha$ curve. For a given significance level $\alpha$, we enforce an upper bound on the FPR at $\alpha$ with a high probability guarantee. Analysis of these figures confirms that our method reliably controls the Type I error, thereby validating the theoretical results presented in Section 2.2. Additional error control analysis including FNR and more FPR results for FACTTEST are available in Appendix D.4.

**Maximizing Accuracy.** Given a significance level $\alpha$, we can determine the threshold $\hat{\tau}_\alpha$ that minimizes the Type II

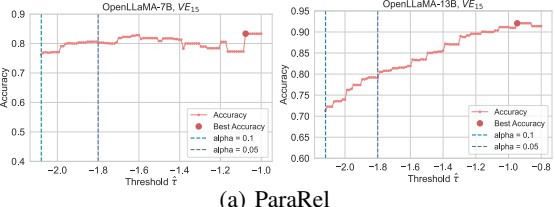

(a) ParaRel

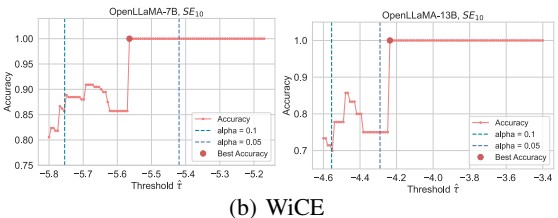

(b) WiCE

Figure 2: The Accuracy-Threshold curve. The title of each sub-figure consists of the dataset name, the model size and the certainty function.

error while ensuring that the Type I error remains within the specified upper bound. For $\hat{\tau} > \hat{\tau}_\alpha$, the Type I error decreases monotonically, whereas the Type II error increases monotonically. Figure 2 presents the accuracy-$\hat{\tau}$ curve, where $\hat{\tau}$ begins at $\hat{\tau}_{0.1}$. This curve can be utilized to maximize accuracy, which does not follow a monotonic trend as the threshold $\hat{\tau}$ increases, while ensuring that the Type I error is controlled below 0.10.

## 5.3. Comparing with Finetuned Models

Figure 3 illustrates the accuracy performance of FACTTEST-t compared to the baseline methods R-Tuning, Finetune-All, and Finetune-Half. We randomly divide $\mathcal{D}$ into two equal parts: $\mathcal{D}_I$ for instruction-tuning and $\mathcal{D}_C$ for constructing the calibration dataset. The pretrained model is finetuned on $\mathcal{D}_I$ to obtain Finetune-Half, while Finetune-All is obtained by training on the entire dataset $\mathcal{D}$. For R-Tuning, we also utilize the entire dataset to finetune the model. It is evident that FACTTEST-t consistently outperforms R-Tuning by a large margin, while utilizing only half of the available training data, thereby reducing training costs by 50%. Notably,

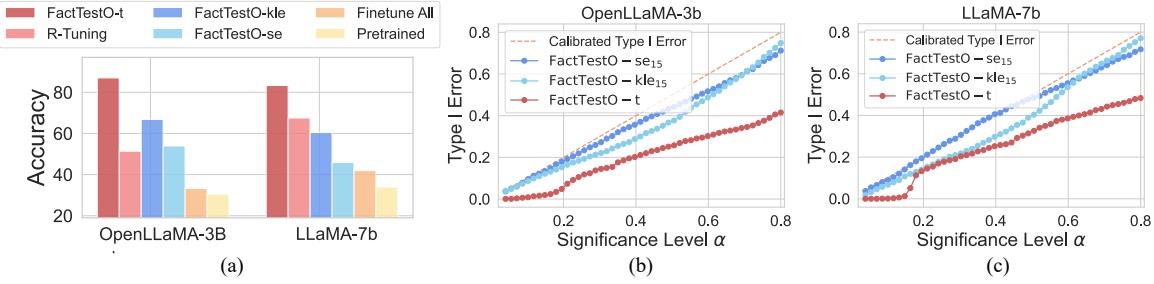

Figure 3: The Accuracy performance (%) of FACTTEST trained on half of the data, comparing with training-based baselines. Both R-Tuning and Finetune All utilize all training data for finetuning, while Finetune Half uses the same half of the finetuning data as FACTTEST.

Figure 4: (a) The Accuracy performance (%) of FACTTEST on ParaRel-OOD testing dataset. (b)(c) FACTTESTO maintains its ability to control Type I error given a significance level $\alpha$ when distribution shifts exist.

FACTTEST-t yields 34% and 28% accuracy improvement over R-Tuning on HotpotQA and FEVER, respectively. Despite the reduced size of the calibration dataset, FACTTEST-t maintains effective control over Type I error, with further details provided in Appendix D.4.

### 5.4. Extension to Covariate Shifts

In this subsection, we evaluate the extension of our framework, denoted as FACTTESTO (FACTTEST for Out-of-distribution domains), on dataset with distribution shifts.

**Setup.** We utilize ParaRel for training, consistent with the aforementioned experiments. We randomly split ParaRel-OOD into a validation dataset comprising 1,000 samples and a testing dataset containing 12k samples. To calculate the density ratio in Section 3 between the target distribution and the source distribution, we employ the training data from the source domain and the validation data from the target domain to train a binary classifier and utilize the predicted probability to approximate density ratios. Subsequently, we select $B$ as the $\gamma$ upper quantile of density ratios to filter out anomalous values. We set the default value of $\gamma$ as 90%.

**Experimental Results.** Figure 4 depicts the performance of FACTTESTO on the ParaRel-OOD testing dataset, alongside the Type I error-$\alpha$ curve. The results demonstrate that FACTTESTO-t significantly outperforms baseline methods by a large margin. Notably, when utilizing OpenLLaMA-3B as the pretrained model, both FACTTESTO-se and FACTTESTO-kle outperform training-based methods with-

Table 3: The accuracy performance (%) of FACTTEST applied to GPT-4o-mini. The significance level is chosen as 5%. The number in parentheses is Type I error. GPT + 7B/13B means utilizing OpenLLaMA-7B/13B to calculate certainty scores for GPT-4o mini.

| Dataset | Model | FTEST-se$_{10}$ | FTEST-se$_{15}$ | FTEST-kle$_{15}$ |
|---------|-------|-----------------|-----------------|------------------|
| ParaRel | GPT-4o mini | 52.83 | | |
| | GPT + 7B | 77.78 (0.03) | 77.31 (0.03) | 83.88 (0.05) |
| | GPT + 13B | 76.91 (0.04) | 77.67 (0.05) | 85.84 (0.04) |
| WiCE | GPT-4o mini | 75.67 | | |
| | GPT + 7B | 81.82 (0.02) | 76.67 (0.03) | – |
| | GPT + 13B | 86.95 (0.01) | 81.77 (0.02) | – |

out fine-tuning. Additionally, FACTTESTO effectively enforces the upper bound on the Type I error, thereby maintaining its efficacy in out-of-distribution scenarios.

### 5.5. Extension to Black-box APIs

We further evaluate our framework on black-box models to broaden the applicability of our framework. While score functions like SE and KLE require token probabilities, which are unavailable for black-box APIs, we utilize open-source models to calculate the scores on calibration datasets constructed by black-box models. Table 3 illustrates the performance of FACTTEST on GPT-4o Mini. The results demonstrate that the scores derived from open-source models are effective for black-box APIs, achieving a 33% accuracy improvement on ParaRel and an 11% improvement on WiCE, while maintaining control over Type I error. These

findings illustrate that our framework provides a practical and effective solution for detecting hallucinations in closed-box models. More results involving Claude-3.5, Gemini-1.5 and GPT-4o are provided in App. D.6.

## 6. Related Work

We review the progress on LLM Factuality and distribution-free and finite-sample inference. More related work is deferred to App. B

**Factuality of LLMs.** The factuality of LLMs is a major problem and of significant research interest (Ji et al., 2023; Maynez et al., 2020a; Li et al., 2023) , including hallucination detection, mitigation and evaluation (Huang et al., 2023; Wang et al., 2023). Our work relates more to hallucination detection, which is imperative for assuring the reliability of the generated content. Kadavath et al. (2022) proposes self-evaluation to verify the prediction. **?** trains a classifier based on hidden layer activations. Lee et al. (2023) uses factual and nonfactual prompts and creates a benchmark for measuring the factuality of generations. Manakul et al. (2023) introduces SelfCheckGPT to fact-check the responses of LLMs in a zero-resource fashion. Zhang et al. (2023) instructs LLMs to refuse unknown questions by refusal-aware instruction tuning. However, none of these works have provided theoretical guarantees.

**Distribution-Free and Finite-Sample Inference.** Recent works have extended conformal prediction to provide guarantees on the outputs of LLMs (Kang et al., 2024; Quach et al., 2024; Mohri & Hashimoto, 2024). While these methods focus on improving model outputs, FACTTEST is designed to verify correctness and decline unknown questions. Our framework leverages principles from Neyman-Pearson (NP) classification. The NP classification paradigm differs from standard classification and cost-sensitive learning, which aim at minimizing a weighted combination of Type I and II errors. Instead, it prioritizes controlling the Type I error while minimizing the Type II error, ensuring that the Type I error remains below a user-specified threshold $\alpha$. Rigollet & Tong (2011) and Scott & Nowak (2005) proposed using empirical risk minimization, and Tong (2013) employed plug-in approaches to construct NP classifiers. A more related work by Tong et al. (2018) introduced an umbrella algorithm that achieves Type I error control for any pretrained classifier, while similar techniques were also proposed in the PAC-style conformal prediction literature (Vovk, 2012). However, the methods in Tong et al. (2018) and Vovk (2012) do not provide Type II error guarantees.

Our work takes an initial step to use the NP classification idea to conduct factuality testing for LLMs. Furthermore, the Type II error analysis of our method can be directly applied to the standard NP umbrella algorithm, which is of independent interest. Additionally, we extend the NP classification framework to account for covariate shifts, enabling it to address more practical, real-world problems.

## 7. Conclusion: Summary and Limitations

In this paper, we introduce FACTTEST that leverages the principles of NP classification to provide statistical guarantees for LLMs. By formulating hallucination detection as a hypothesis testing problem, FACTTEST enforces an upper bound on Type I errors. It ensures strong power control under mild conditions and can be extended to maintain its effectiveness in the presence of covariate shifts. These theoretical analyses can be seamlessly integrated with the standard NP umbrella algorithm, not limited to our framework. Our approach is distribution-free and finite-sample. It applies to any LLM including closed-box models. Empirical evaluations have demonstrated that FACTTEST consistently outperforms baselines and maintains superior performance under distribution shifts. Additionally, our framework effectively enhances the reliability of black-box APIs.

One limitation of our work is the implementation of a limited number of score functions. Exploring additional score functions could further enhance the performance. Furthermore, our framework constructs the predictor in an offline manner. Future work could extend FACTTEST to support online testing, thus enabling real-time factuality assessments.

## Acknowledgements

The research of LZ is partially supported by NSF CAREER DMS-2340241.

## Impact Statement

Our work aims to enhance the reliability of large language models (LLMs) by statistically evaluating the correctness of model outputs. The primary objective is to reduce the risk of misinformation and non-factual content, especially in high-stakes domains where inaccuracies can lead to significant harm. FACTTEST facilitates responsible deployment of LLMs by enabling them to decline to answer uncertain questions, thereby promoting transparency and trustworthiness in AI systems. While the framework is not directly tied to specific ethical risks, it indirectly mitigates the propagation of false or misleading information in sensitive domains. Additionally, its extensibility to black-box APIs supports the development of more robust and fair AI ecosystems. Furthermore, we have taken steps to ensure that our research minimizes any privacy or security risks. All datasets used in this study are publicly available and do not contain personally identifiable information, thereby safeguarding user privacy and adhering to data security standards.

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

## A. Derivation and Proof

*Proof of Equation 4.* Assume that when $(q, M(q)) \sim P_0$, $\hat{\eta}(q, M(q))$ has CDF $F$. We denote the $1 - \alpha$ quantile of $F$ as $F^{-1}(1 - \alpha) = \inf\{x \in \mathbb{R} \mid F(x) \geq 1 - \alpha\}$. Then we can show that:

$$\mathbb{P}_{\mathcal{D}}(\mathbb{P}_{(q,M(q))\sim P_0}(\hat{\eta}(q, M(q)) > T_{(k)}) > \alpha) = \mathbb{P}_{\mathcal{D}}(T_{(k)} < F^{-1}(1 - \alpha)) \tag{10}$$

Considering the property of the order statistics, we have that

$$\mathbb{P}_{\mathcal{D}}(T_{(k)} < F^{-1}(1 - \alpha)) = \mathbb{P}_{\mathcal{D}}\left(\sum_{i=1}^{n_0} \mathbb{I}\left(\hat{\eta}(q_i^{(0)}, M(q_i^{(0)})) < F^{-1}(1 - \alpha)\right) \geq k\right) \tag{11}$$

$$\leq \sum_{j=k}^{n_0} \binom{n_0}{j}(1 - \alpha)^j \alpha^{n-j} \triangleq v(k) \tag{12}$$

where $\mathbb{I}(\cdot)$ is the indicator function, defined as:

$$\mathbb{I}(\hat{\eta}(q_i^{(0)}, M(q_i^{(0)})) < Q_\alpha) = \begin{cases} 1 & \text{if } \hat{\eta}(q_i^{(0)}, M(q_i^{(0)})) > Q_\alpha \\ 0 & \text{otherwise} \end{cases}$$

$\square$

*Proof of Theorem 2.1.* Theorem 2.1 follows from the definition of $\hat{k}$. $\square$

*Lemma 2* (Theorem 1 in (Skorski, 2023)). Suppose $X_1, \ldots, X_n$ are i.i.d. continuous random variables with CDF $F$, denote

$$\epsilon_k = \frac{4(n - 2k + 1)}{3(n + 1)(n + 3)} \log \frac{2}{\delta} \vee 0 + \sqrt{\frac{2k(n - k + 1)}{(n + 1)^2(n + 2)} \log \frac{2}{\delta}},$$

then

$$\mathbb{P}\left(F(X_{(k)}) - \frac{k}{n + 1} \leq \epsilon_k\right) \geq 1 - \delta.$$

*Proof of Theorem 2.2.* At first, to simplify the notations in the proof, we argue that the function $H$ can be assumed to be identity without the loss of generality. To see this, note that $\hat{k}$ does not depend on the choice of $\hat{\eta}$ and $\mathbb{I}(\hat{\eta} > T_{(\hat{k})}) = \mathbb{I}(H \circ \hat{\eta} > (H(T))_{(\hat{k})})$ with $(H(T))_{(\hat{k})}$ to be the $k$-th smallest order statistic of $\{H(T_i) : i \in [n_0]\}$, therefore, $\hat{f}_\alpha$ is invariant if we replace $\hat{\eta}$ in Section 2 by $H \circ \hat{\eta}$. Consequently, without the loss of generality, we assume $H$ is the identity function. Then the proof of Theorem 2.2 consists of three parts.

1) Firstly, we show that $\mathcal{R}_0(\hat{f}_\alpha)$ is not much smaller than $\alpha$. To see this, since we assume $\hat{\eta}(q, M(q))$ with $(q, M(q)) \sim P_0$ is a continuous random variable, it follows from the definition of $\hat{k}$ that

$$\mathbb{P}(F(T_{(\hat{k}-1)}) < 1 - \alpha) = \mathbb{P}(\mathbb{P}_{(q,M(q))\sim P_0}(\hat{\eta}(q, M(q)) > T_{(\hat{k}-1)}) > \alpha) > \delta.$$

Here we only consider the case where $\hat{k} > 1$, as will be shown in Equation (13), it holds as long as $n_0$ is not too small. Since $\hat{k}$ is deterministic given $n_0$, it follows from Lemma 2 that

$$\mathbb{P}\left(F(T_{(\hat{k}-1)}) \leq \frac{\hat{k} - 1}{n_0 + 1} - \epsilon_{\hat{k}-1}\right) \leq \delta.$$

Therefore we have

$$\frac{\hat{k} - 1}{n_0 + 1} - \epsilon_{\hat{k}-1} < 1 - \alpha.$$

Denote the event $E_1$ as

$$E_1 = \left\{F(T_{(\hat{k})}) \leq \frac{\hat{k}}{n_0 + 1} + \epsilon_{\hat{k}}\right\},$$

it follows from Lemma 2 that $\mathbb{P}(E_1) \geq 1 - \delta$. Under $E_1$, we know

$$F(T_{(\hat{k})}) \leq \frac{\hat{k}}{n_0 + 1} + \epsilon_{\hat{k}} < 1 - \alpha + \frac{1}{n_0 + 1} + \epsilon_{\hat{k}-1} + \epsilon_{\hat{k}},$$

which implies

$$\mathbb{P}_{(q,M(q))\sim P_0}(\hat{\eta}(q, M(q)) > T_{(\hat{k})}) = 1 - F(T_{(\hat{k})}) > \alpha - \frac{1}{n_0 + 1} - \epsilon_{\hat{k}-1} - \epsilon_{\hat{k}}.$$

Similarly, since

$$\mathbb{P}(F(T_{(\hat{k})}) \geq 1 - \alpha) \geq 1 - \delta, \quad \mathbb{P}\left(F(T_{(\hat{k})}) \leq \frac{\hat{k}}{n_0 + 1} + \epsilon_{\hat{k}}\right) \geq 1 - \delta,$$

we know

$$1 - \alpha \leq \frac{\hat{k}}{n_0 + 1} + \epsilon_{\hat{k}},$$

which concludes that

$$\hat{k} \gtrsim (1 - \alpha)n_0. \tag{13}$$

Thus we have

$$\epsilon_{\hat{k}-1} + \epsilon_{\hat{k}} \lesssim \sqrt{\frac{\alpha}{n_0} \log \frac{1}{\delta}},$$

and

$$\mathbb{P}_{(q,M(q))\sim P_0}(\hat{\eta}(q, M(q)) > T_{(\hat{k})}) > \alpha - c\sqrt{\frac{\alpha}{n_0} \log \frac{1}{\delta}}. \tag{14}$$

2) Secondly, we show $T_{(\hat{k})}$ is close to $\tau_\alpha$. Denote $\alpha' = \alpha - c\sqrt{\frac{\alpha}{n_0} \log \frac{1}{\delta}}$, it follows from Equation (14) that under $E_1$, we have

$$\mathbb{P}_{(q,M(q))\sim P_0}(\hat{\eta}(q, M(q)) > T_{(\hat{k})}) > \alpha' = \mathbb{P}_{(q,M(q))\sim P_0}(\eta(q, M(q)) > \tau_{\alpha'})$$
$$\geq \mathbb{P}_{(q,M(q))\sim P_0}(\hat{\eta}(q, M(q)) > \tau_{\alpha'} + \epsilon_\eta),$$

so

$$T_{(\hat{k})} < \tau_{\alpha'} + \epsilon_\eta.$$

Denote $E_2$ as

$$E_2 = \{\mathbb{P}_{(q,M(q))\sim P_0}(\hat{\eta}(q, M(q)) > T_{(\hat{k})}) \leq \alpha\},$$

we know $\mathbb{P}(E_2) \geq 1 - \delta$. Under $E_2$, we have

$$\mathbb{P}_{(q,M(q))\sim P_0}(\hat{\eta}(q, M(q)) > T_{(\hat{k})}) \leq \alpha = \mathbb{P}_{(q,M(q))\sim P_0}(\eta(q, M(q)) > \tau_\alpha)$$
$$\leq \mathbb{P}_{(q,M(q))\sim P_0}(\hat{\eta}(q, M(q)) > \tau_\alpha - \epsilon_\eta),$$

so

$$T_{(\hat{k})} \geq \tau_\alpha - \epsilon_\eta.$$

Then we conclude that

$$|T_{(\hat{k})} - \tau_\alpha| \leq \tau_{\alpha'} - \tau_\alpha + \epsilon_\eta = \epsilon_\tau.$$

3) Now we are able to control the excess risk of $\hat{f}_\alpha$. For any $(q, M(q))$, if we use $Y = 0$ (resp. $Y = 1$) to denote the model $M$ is uncertain (resp. certain) of $q$, and denote $p_y = \mathbb{P}(Y = 1)$ to be the marginal distribution for $M$ to be certain, then

$$\frac{dP_1}{dP_0}(q, M(q)) = \frac{\eta(q, M(q))(1 - p_y)}{(1 - \eta(q, M(q)))p_y},$$

which is increasing in $\eta(q, M(q))$. Denote $\xi_\alpha = \frac{\tau_\alpha(1-p_y)}{(1-\tau_\alpha)p_y}$, if $|\eta(q, M(q)) - \tau_\alpha| \leq \epsilon_\tau + \epsilon_\eta$ and $\tau_\alpha + \epsilon_\tau + \epsilon_\eta < 1$, then

$$\left|\frac{dP_1}{dP_0}(q, M(q)) - \xi_\alpha\right| \leq \frac{(1 - p_y)(\epsilon_\tau + \epsilon_\eta)}{p_y(1 - \tau_\alpha - \epsilon_\tau - \epsilon_\eta)^2}.$$

Then under $E_1 \cap E_2$, we can control the excess risk as

$$\mathcal{R}_1(\hat{f}_\alpha) - \mathcal{R}_1(f_\alpha^*)$$

$$=\mathbb{E}_{(q,M(q))\sim P_0} \frac{dP_1}{dP_0}(q, M(q))\Big(\mathbb{I}(\hat{\eta}(q, M(q)) \le T_{(\hat{k})}) - \mathbb{I}(\eta(q, M(q)) \le \tau_\alpha)\Big)$$

$$=\mathbb{E}_{(q,M(q))\sim P_0} \left(\frac{dP_1}{dP_0}(q, M(q)) - \xi_\alpha\right)\Big(\mathbb{I}(\hat{\eta}(q, M(q)) \le T_{(\hat{k})}) - \mathbb{I}(\eta(q, M(q)) \le \tau_\alpha)\Big)$$

$$+ \xi_\alpha \mathbb{E}_{(q,M(q))\sim P_0}\Big(\mathbb{I}(\hat{\eta}(q, M(q)) \le T_{(\hat{k})}) - \mathbb{I}(\eta(q, M(q)) \le \tau_\alpha)\Big)$$

$$=\mathbb{E}_{(q,M(q))\sim P_0} \left|\frac{dP_1}{dP_0}(q, M(q)) - \xi_\alpha\right|\left|\mathbb{I}(\hat{\eta}(q, M(q)) \le T_{(\hat{k})}) - \mathbb{I}(\eta(q, M(q)) \le \tau_\alpha)\right|$$

$$+ \xi_\alpha \Big(\mathcal{R}_0(f_\alpha^*) - \mathcal{R}_0(\hat{f}_\alpha)\Big)$$

$$\le \mathbb{E}_{(q,M(q))\sim P_0} \left|\frac{dP_1}{dP_0}(q, M(q)) - \xi_\alpha\right|\mathbb{I}\Big(|\eta(q, M(q)) - \tau_\alpha| \le \epsilon_\tau + \epsilon_\eta\Big) + c\xi_\alpha\sqrt{\frac{\alpha}{n_0}\log\frac{1}{\delta}}$$

$$\lesssim \frac{(1 - p_y)(\epsilon_\tau + \epsilon_\eta)}{p_y(1 - \tau_\alpha - \epsilon_\tau - \epsilon_\eta)^2}G_\alpha(\epsilon_\tau + \epsilon_\eta) + \xi_\alpha\sqrt{\frac{\alpha}{n_0}\log\frac{1}{\delta}}.$$

$\square$

*Proof of Theorem 3.1.* Note that once we show $\tilde{\mathcal{D}}_0 \mid \mathcal{I} \stackrel{\text{i.i.d.}}{\sim} P_0$, then similar to Theorem 2.1,

$$\mathbb{P}_\mathcal{D}(\mathbb{P}_{(q,M(q))\sim P_0}(\hat{f}_\alpha(q, M(q)) = 1) \le \alpha \mid \mathcal{I}) \ge 1 - \delta.$$

Taking expectation with respect to $\mathcal{I}$ concludes the result.

It remains to prove $\tilde{\mathcal{D}}_0 \mid \mathcal{I} \stackrel{\text{i.i.d.}}{\sim} P_0$. To this end, it suffices to show $(q, M(q))|\{U \le w(q, M(q))\} \sim P_{q,M(q)|y=0} = P_0$ with $(q, M(q)) \sim \tilde{P}_0$. For any measurable set $C \subset \mathcal{Q} \times \mathcal{A}$, the conditional distribution of $(q, M(q))|\{U \le w(q, M(q))\}$ can be expressed as

$$\mathbb{P}\big((q, M(q)) \in C | U \le w(q, M(q))\big)$$

$$=\frac{\mathbb{P}\big((q, M(q)) \in C, U \le w(q, M(q))\big)}{\mathbb{P}\big(U \le w(q, M(q))\big)}$$

$$=\frac{\mathbb{E}\frac{w(q,M(q))}{B}\mathbb{I}\big((q, M(q)) \in C\big)}{\mathbb{E}\frac{w(q,M(q))}{B}}$$

$$=\mathbb{P}_{(q,M(q))\sim P_{q,M(q)|y=0}}\big((q, M(q)) \in C\big),$$

where we have use the facts that $\mathbb{P}(U \le w(q, M(q))|q, M(q)) = \frac{w(q,M(q))}{B}$, $\mathbb{E}_{(q,M(q))\sim \tilde{P}_0}w(q, M(q)) = 1$ and $\mathbb{E}_{(q,M(q))\sim \tilde{P}_0}w(q, M(q))\mathbb{I}((q, M(q)) \in C) = \mathbb{P}_{(q,M(q))\sim P_0}((q, M(q)) \in C)$. $\square$

# B. More Related Works

**Uncertainty Quantification and Confidence Calibration.** Our work relates to a line of work on uncertainty quantification (UQ) for LLMs, as we can employ these functions to assess models' ability to reliably give an answer. Predictive entropy that measures the entropy of the model's predicted token distribution has been used as a simple baseline for UQ in LLMs (Braverman et al., 2019). Kuhn et al. (2023) introduced Semantic Entropy, which incorporates linguistic invariances to measure uncertainty. Most recently, Nikitin et al. (2024) introduced Kernel Language Entropy (KLE), which defines positive semidefinite unit trace kernels and quantifies uncertainty using the von Neumann entropy. Lin et al. (2024) identifies that existing methods treat all tokens equally when estimating uncertainty and proposed a simple supervised approach for uncertainty estimation in black-box LLMs using labeled datasets. Duan et al. (2024) proposes jointly shifting attention to more relevant (SAR) components. Besides, recent research on confidence calibration for LLMs has explored several innovative approaches. For example, Tian et al. (2023) elicits verbalized confidences to improve calibration. Huang et al. (2024) proposes confidence elicitation methods for long-form generations. Multicalibration (Detommaso et al., 2024) aims to ensure LLM confidence scores accurately reflect the true likelihood of predictions being correct. These works are

complementary to ours, as our contribution is a meta-algorithm that works with any uncertainty quantification method to serve as score functions. Future developments in this line of work can greatly improve the performance of our framework.

**Conformal Prediction.** Conformal prediction enables the construction of confidence sets that contain the true outcome with a specified probability (Shafer & Vovk, 2007; Angelopoulos & Bates, 2022; Barber et al., 2023). It has been successfully applied to various black-box machine learning models (Angelopoulos & Bates, 2022) but has limited application in language models (LMs). Specifically, Kumar et al. (2023) provides conformal guarantees on multiple-choice datasets. C-RAG (Kang et al., 2024) provides conformal risk analysis for RAG models and certifies an upper confidence bound. CLM (Quach et al., 2024) extends conformal prediction for LLM generations and provide coverage guarantees. TRAQ (Li et al., 2024) uses conformal prediction to construct prediction sets both for retrieved sets and prediction sets to provide guarantees for RAG. Conformal Factuality (Mohri & Hashimoto, 2024) enables the application of conformal prediction in improving model performance. However, traditional CP methods in LLMs focus primarily on coverage guarantees without differentiating between correct and incorrect samples, thereby lacking explicit error rate controls essential for hallucination detection. In contrast, FACTTEST differs from those works in that it aims to evaluate the model's ability to answer correctly, detect hallucinations and explicitly control both Type I and Type II errors.

# C. Implementation Details

## C.1. Details about Baselines

We provide a detailed explanation of the settings used for the baseline.

- **SAPLMA**: We follow the original settings and implement SAPLMA (Azaria & Mitchell, 2023) to serve as a training-free baseline. SAPLMA trains a classifier to predict statement truthfulness. We evaluate models on questions that the model's answer is considered as truthful. The corresponding results are shown in Appendix D.5.
- **SelfCheckGPT**: We implement SelfCheckGPT with NLI (Manakul et al., 2023) as recommended by the authors. It utilizes Natural Language Inferencce (NLI) model to predict entailment or contradiction. We sample five answers and the model will output a probability of contradiction from 0 to 1. We evaluate the base models on questions with the contradiction score less than 0.5.
- **R-Tuning**: We follow the settings in the original paper (Zhang et al., 2023): We first construct a refusal-aware dataset by adding prompt *'Are you sure you accurately answered the question based on your internal knowledge?'* and the corresponding *'Sure'* or *'Unsure'* to each question-answer pair, and then finetune the model on this dataset. We evaluate the finetuned model on questions that the model is certain.

## C.2. Details about Datasets

We conduct our experiments on four datasets and follow the same train/test split in Zhang et al. (2023), which are described as follows.

- **ParaRel** (Elazar et al., 2021): This dataset comprises factual knowledge with various prompts and relations initially designed for mask prediction. It is utilized to evaluate the model's ability to comprehend paraphrased relational facts. To adapt ParaRel for autoregressive models, Zhang et al. (2023) reformatted it into a question-answering format. Duplicate prompts with different templates but identical entities were removed for our question-answering task, resulting in 25,133 prompt-answer pairs across 31 domains. Zhang et al. (2023) divided ParaRel into two subsets: the first 15 domains serve as in-domain data, and the remaining 16 domains as out-of-domain data (13974 samples). The in-domain data is further split equally into training and testing sets, consisting of 5575 and 5584 samples.
- **HotpotQA** (Yang et al., 2018): It is a question-answering dataset that necessitates complex reasoning across multiple documents. We evaluate the model by providing the relevant context documents and questions to assess its ability to generate correct answers. The development set is used as the testing set for our evaluations. The training set contains 10K samples randomly selected from the original dataset while the testing set contains 7405 samples.
- **WiCE** (Kamoi et al., 2023): It is a natural language inference (NLI) dataset focused on textual entailment. Each data sample consists of an evidence statement and a claim, and the model must determine whether the evidence supports, partially supports, or does not support the claim. We utilize this dataset as multiple-choice questions with three options for each question. The training and testing sets contain 3470 and 958 samples, respectively.
- **FEVER** (Thorne et al., 2018a): FEVER consists of claims paired with supporting evidence from Wikipedia. Each claim is classified as SUPPORTED, REFUTED, or NOT ENOUGH INFO. This dataset is employed to assess the

models' capability to verify the factual accuracy of statements against Wikipedia documents. We utilize FEVER as a multiple-choice NLI task with three options for each question: (A) SUPPORTED, (B) REFUTED, (C) NOT ENOUGH INFO. The training set contains 10K samples randomly selected from the original dataset while the testing set contains 9999 samples.

Detailed information about the original datasets and the data preprocessing procedures can be found in Zhang et al. (2023). In Figure 5, we illustrate the distribution of correct and incorrect data within the constructed datasets $D_0, D_1$.

### C.3. Details about Score Functions

We implement our framework with three entropy-based certainty functions. Details are described as follows.

- **Vanilla Entropy:** The frequency of a predicted answer $M(q)_j$ in Equation. 8 is calculated by $\frac{m}{k}$, where $m$ is the number of times $M(q)_j$ exists in $k$ generations.

- **Semantic Entropy**: Semantic entropy is an entropy which incorporates linguistic invariances created by shared meanings (Kuhn et al., 2023), which is computed by the probability distribution over meanings.

$$SE(q, M(q)) = -\sum_c p(c|q) \log p(c|q) = -\sum_c \left( \left( \sum_{\in c} p(|\ q) \right) \log \left[ \sum_{\in c} p(|\ q) \right] \right) \quad (15)$$

where $c$ represents possible meaning-class and $p(|q)$ is the probability of the entire answer sequence, that is, the product of the conditional probabilities of new tokens given past tokens. This can be approximated by:

$$SE(q, M(q)) \approx -|C|^{-1} \sum_{i=1}^{|C|} \log p(C_i \mid q), \hat{\eta}(q, M(q)) = -SE(q, M(q)). \quad (16)$$

We follow (Kuhn et al., 2023) to estimate the expectation of 16 given that we cannot have access to all possible $c$. We query $M$ $k$ times and divide the answers into semantic classes $C$ based on semantic equivalence.

Notably, for multiple-choice datasets including WiCE and FEVER, the outputs are among three choices. In this case, we view different tokens as having different semantic meanings, and the semantic entropy is thus reduced to predictive entropy.

- **Kernel Language Entropy**: Kernel language entropy (KLE) is a generalization of semantic entropy (Nikitin et al., 2024), providing more detailed uncertainty estimates by considering pairwise semantic dependencies between answers or semantic clusters. It quantifies uncertainty by constructing a semantic kernel from the model's $k$ generated answers and computing its von Neumann entropy. Specifically, for a given input $q$, we generate $k$ responses, build a positive semidefinite semantic kernel $K_{\text{sem}}$ that captures the semantic relationships among these answers, and then calculate the von Neumann entropy (VNE) of this kernel. The KLE can be defined as:

$$\text{KLE}(q, M(q)) = \text{VNE}(K_{sem}) = -[K_{\text{sem}} \log K_{\text{sem}}], \ \hat{\eta}(q, M(q)) = -KLE(q, M(q)). \quad (17)$$

where, $K_{sem}$ is the semantic kernel which can be implemented from semantic graphs over the LLM outputs.

### C.4. Details about FACTTESTO

In order to approximate the density ratio, we randomly split 1000 samples from ParaRel-OOD as validation samples and the remaining 12k samples as testing samples. We then utilize the supervised identification strategy to divide the validation samples into $\mathcal{D}_0'$ and $\mathcal{D}_1'$, and the training dataset into $\mathcal{D}_0$ and $\mathcal{D}_1$. We extract the features from the questions in $\mathcal{D}_0'$, $\mathcal{D}_0$ by a *TfidfVectorizer*, and label them as 1 (target data) and 0 (source data). We then utilize logistic regression to train a binary classifer and use the predicted probability to approximate density ratios.

## D. More Experiemnt Results

In this section, we provide more experiment results, including experiments with more significance levels, answer rate analysis, more error control analysis, experiments with more base models and certainty distribution visualizations.

## D.1. Ablation Study

As is demonstrated in Sec. 4, FACTTEST can also be formulated with PAC-style conformal prediction (CP) conditioned on $y = 0$. As illustrated in Sec. 4, directly checking whether outputs fall within the conformal prediction set does not work for hallucination detection. In this part, we implement two variants with conformal prediction that could control Type I errors to some extent and compare them with FACTTEST to further illustrate the efficacy of our method. Unlike previous works that consider prediction sets for LLM generated outputs, we consider conformal prediction sets around a binary classifier that takes $X = (q, M(q))$ as inputs and outputs whether the generated answer is correct ($y = 1$) or incorrect ($y = 0$). The model will abstain from answering the question if $y = 0$ is included in the prediction set. We then set the non-conformity score $s(X, y)$ to be $\text{Sigmoid}(\hat{\eta})$ when $y = 0$ and $1 - \text{Sigmoid}(\hat{\eta})$ when $y = 1$.

- **Standard Conformal Prediction.** The prediction set satisfies marginal coverage:

$$\mathbb{P}(y_{\text{test}} \in \mathcal{C}_{\text{Standard}}(X_{\text{test}})) \geq 1 - \alpha.$$

The standard conformal prediction set is given by:

$$\mathcal{C}_{\text{Standard}}(X_{\text{test}}) = \{y : s(X_{\text{test}}, y) \leq \hat{q}\},$$

where

$$\hat{q} = \text{Quantile}\left(\left\lceil \frac{(N+1)(1-\alpha)}{N} \right\rceil, \{s_i\}_{i=1}^N\right)$$

- **Class-conditional Conformal Prediction.** The prediction set satisfies class-conditional coverage:

$$\mathbb{P}(y_{\text{test}} \in \mathcal{C}_{\text{Classwise}}(X_{\text{test}}) \mid y_{\text{test}} = 0) \geq 1 - \alpha$$

Suppose there are $n_0$ samples where $y = 0$. The class-conditional conformal prediction set is given by:

$$\mathcal{C}_{\text{Classwise}}(X_{\text{test}}) = \{y : s(X_{\text{test}}, y) \leq \hat{q}^{(0)}\},$$

where

$$\hat{q}^{(0)} = \text{Quantile}\left(\left\lceil \frac{(n_0+1)(1-\alpha)}{n_0} \right\rceil, \{s_i\}_{i=1}^{n_0}\right)$$

Table 4: The accuracy performance (%) of FACTTEST on ParaRel. The $\alpha$ for FACTTEST and two variants are all set to 0.05. The percentages inside the parentheses are the Type I error. We use 15-generation SE as score function.

| Dataset | Base Model | Class-Conditional CP | FACTTEST-se$_{15}$ |
|---|---|---|---|
| ParaRel | OpenLLAMA-3B | 65.54 (0.05) | 67.26 (0.05) |
| | OpenLLaMA-7B | 63.20 (0.05) | 65.23 (0.04) |
| | OpenLLaMA-13B | 71.45 (0.05) | 73.09 (0.03) |
| FEVER | OpenLLAMA-3B | 83.17 (0.06) | 83.90 (0.05) |
| | LLaMA-7B | 31.71 (0.06) | 33.27 (0.05) |
| | LLaMA-13B | 50.41 (0.07) | 52.23 (0.05) |

**Results.** Standard CP leverages the full calibration dataset $\mathcal{D}$ while Class-conditional CP as well as FACTTEST exclusively utilize $\mathcal{D}_0$. We observe that standard CP exhibits excessive conservatism due to its requirement to satisfy coverage guarantees for both $y = 0$ and $y = 1$ simultaneously. This often results in rejection of all the questions, leading to 0% predictive accuracy in practice. Table. 4 demonstrates the accuracy and Type I error performance of FACTTEST compared with class-conditional CP variant. Class-conditional CP aims at controlling the average type I error, while FACTTEST enforces stricter control by bounding the $\delta$-quantile of all type I errors. Therefore, FACTTEST provides statistically more stringent guarantees than class-conditional CP, as demonstrated by its lower empirical Type I error rates while maintaining competitive accuracy.

## D.2. More significance levels.

Table 5 presents the accuracy performance of FACTTEST in comparison with base pretrained models at a significance level of $\alpha = 0.10$. Similarly, Table 6 reports the corresponding Type I error rates under the same significance level. The results

Table 5: The accuracy performance (%) of FACTTEST compared to Pretrained models on question-answering and multiple-choice datasets using a significance level of $\alpha = 0.10$.

| Dataset | Model | Pretrained | FTEST-ve$_5$ | FTEST-ve$_{10}$ | FTEST-ve$_{15}$ | FTEST-se$_5$ | FTEST-se$_{10}$ | FTEST-se$_{15}$ | FTEST-kle$_{15}$ |
|---|---|---|---|---|---|---|---|---|---|
| ParaRel | OpenLLaMA-3B | 36.66 | 60.54 | 67.22 | 67.10 | 61.01 | 62.32 | 63.05 | **75.51** |
| | OpenLLaMA-7B | 40.38 | 74.92 | **77.84** | 76.89 | 69.00 | 68.78 | 64.97 | 75.36 |
| | OpenLLaMA-13B | 42.21 | 77.37 | 75.17 | 79.25 | 68.93 | 68.45 | 68.82 | **79.55** |
| HotpotQA | OpenLLaMA-3B | 25.72 | 50.81 | 49.84 | 51.68 | 45.41 | 45.23 | 46.88 | **52.70** |
| | OpenLLaMA-7B | 28.63 | 56.06 | 56.23 | 55.77 | 51.10 | 51.63 | 52.33 | **56.73** |
| | LLaMA-13B | 30.83 | 51.49 | 51.45 | 51.61 | 53.42 | 53.12 | **55.38** | 53.34 |
| WiCE | OpenLLaMA-3B | 64.72 | 67.65 | 64.40 | **76.27** | 61.54 | 64.71 | 64.86 | – |
| | OpenLLaMA-7B | 72.96 | 50.00 | 63.77 | 57.32 | 83.33 | **85.71** | 74.19 | – |
| | LLaMA-13B | 56.89 | 63.33 | 50.00 | 57.14 | 75.00 | 67.44 | **77.42** | – |
| FEVER | OpenLLaMA-3B | 39.74 | 60.24 | 53.06 | 52.00 | 80.71 | 82.00 | **82.29** | – |
| | LLaMA-7B | 35.99 | 43.92 | 43.33 | **47.73** | 28.69 | 31.49 | 32.82 | – |
| | LLaMA-13B | 32.15 | 38.74 | 42.48 | 46.79 | 51.95 | **53.01** | 50.92 | – |

Table 6: The Type I error of FACTTEST on question-answering and multiple-choice datasets, with a significance level $\alpha = 0.10$.

| Dataset | Model | FTEST-ve$_5$ | FTEST-ve$_{10}$ | FTEST-ve$_{15}$ | FTEST-se$_5$ | FTEST-se$_{10}$ | FTEST-se$_{15}$ | FTEST-kle$_{15}$ |
|---|---|---|---|---|---|---|---|---|
| ParaRel | OpenLLaMA-3B | 0.0455 | 0.0732 | 0.0783 | 0.0851 | 0.0865 | 0.0905 | 0.0795 |
| | OpenLLaMA-7B | 0.0225 | 0.0240 | 0.0348 | 0.0799 | 0.0886 | 0.0829 | 0.0781 |
| | OpenLLaMA-13B | 0.0192 | 0.0226 | 0.0325 | 0.0849 | 0.0706 | 0.0589 | 0.0709 |
| HotpotQA | OpenLLaMA-3B | 0.0276 | 0.0585 | 0.0521 | 0.0660 | 0.0678 | 0.0651 | 0.0605 |
| | OpenLLaMA-7B | 0.0295 | 0.0597 | 0.0637 | 0.0631 | 0.0643 | 0.0616 | 0.0590 |
| | LLaMA-13B | 0.0222 | 0.0556 | 0.0675 | 0.0611 | 0.0675 | 0.0503 | 0.0667 |
| WiCE | OpenLLaMA-3B | 0.0325 | 0.0621 | 0.0414 | 0.0443 | 0.0355 | 0.0325 | – |
| | OpenLLaMA-7B | 0.0694 | 0.0965 | 0.1151 | 0.0347 | 0.0154 | 0.0308 | – |
| | LLaMA-13B | 0.0266 | 0.0532 | 0.0799 | 0.0169 | 0.0338 | 0.0169 | – |
| FEVER | OpenLLaMA-3B | 0.0164 | 0.0418 | 0.0600 | 0.1039 | 0.1053 | 0.1042 | – |
| | LLaMA-7B | 0.0598 | 0.0617 | 0.0556 | 0.0928 | 0.1027 | 0.1091 | – |
| | LLaMA-13B | 0.0172 | 0.0828 | 0.0709 | 0.0944 | 0.1059 | 0.1136 | – |

Table 7: The answer rate and accuracy performance (%) of FACTTEST-t. The number in parenthese is Answer Rate, which means the percentage of willingly answered questions.

| Dataset | Model | Finetuned | R-Tuning | FTEST-t ($\alpha = 0.15$) | FTEST-t ($\alpha = 0.10$) | FTEST-t ($\alpha = 0.05$) |
|---|---|---|---|---|---|---|
| ParaRel | OpenLLaMA-3B | 61.73 ( 100% ) | 87.42 ( 37% ) | 89.91 ( 46% ) | 92.73 ( 31% ) | 94.26 ( 17% ) |
| | LLaMA-7B | 67.73 ( 100% ) | 89.65 ( 42% ) | 92.76 ( 47% ) | 95.04 ( 31% ) | 96.01 ( 18% ) |
| FEVER | OpenLLaMA-3B | 65.56 ( 100% ) | 67.19 ( 11% ) | 92.58 ( 38% ) | 94.88 ( 36% ) | 97.82 ( 33% ) |
| | LLaMA-7B | 66.24 ( 100% ) | 66.19 ( 49% ) | 95.41 ( 28% ) | 95.83 ( 24% ) | 96.79 ( 16% ) |

show that Type I error remains effectively controlled with the adjusted $\alpha$. While the accuracies at $\alpha = 0.10$ are slightly lower than those at $\alpha = 0.05$, FACTTEST continues to significantly outperform base models and maintains a lower Type II error rate.

## D.3. Answer Rate Analysis

Table 7 presents the answer rate and corresponding accuracy performance (%) of FACTTEST-t in comparison with baseline methods across multiple datasets and models. The findings demonstrate that FACTTEST-t consistently achieves higher accuracy while effectively managing the answer rate through varying significance levels ($\alpha$). Specifically, FACTTEST-t with $\alpha = 0.15$ answers 47% questions on ParaRel and acheives 92.76% accuracy, outperforming R-Tuning, which answers 42% of the questions with an accuracy of 89.65%. Similarly, FACTTEST-t maintains superior accuracy performance on FEVER compared to baseline models while managing the answer rate through different significance levels.

Table 8: The accuracy performance (%) of FACTTEST compared to baselines on question-answering and multiple-choice datasets using a significance level of $\alpha = 0.05$. For brevity, FACTTEST is abbreviated as FTEST.

| Dataset | Model | Base | SelfCheckGPT | SAPLMA | FTEST-ve$_{15}$ | FTEST-se$_{15}$ | FTEST-kle$_{15}$ | FTEST-saplma |
|---------|-------|------|--------------|--------|---------|---------|----------|--------------|
| ParaRel | OpenLLaMA-3B | 36.66 | 53.60 | 67.33 | 67.28 | 67.26 | 78.45 | **84.77** |
| | OpenLLaMA-7B | 40.38 | 60.05 | 76.18 | 80.29 | 65.23 | 76.83 | **88.99** |
| | OpenLLaMA-13B | 42.21 | 59.62 | 69.62 | 79.41 | 73.09 | 83.84 | **88.16** |
| HotpotQA | OpenLLaMA-3B | 25.72 | 36.42 | 25.13 | 53.75 | 52.66 | **55.35** | 31.96 |
| | OpenLLaMA-7B | 28.63 | 39.16 | 24.00 | **60.67** | 56.56 | 60.66 | 31.96 |
| | LLaMA-13B | 30.83 | 41.78 | 38.91 | 49.74 | **60.69** | 54.49 | 38.69 |
| WiCE | OpenLLaMA-3B | 64.72 | 66.36 | 64.40 | **68.18** | 66.67 | – | 65.67 |
| | OpenLLaMA-7B | 72.96 | 75.00 | 72.50 | 47.37 | **90.00** | – | 73.33 |
| | LLaMA-13B | 56.89 | 57.39 | 57.67 | 44.44 | **90.00** | – | 41.67 |
| FEVER | OpenLLaMA-3B | 39.74 | 41.97 | **93.21** | 41.72 | **83.90** | – | **93.21** |
| | LLaMA-7B | 35.99 | 40.89 | **74.05** | 51.38 | 33.27 | – | 63.88 |
| | LLaMA-13B | 32.15 | 41.25 | - | 46.07 | **52.23** | – | 49.81 |

## D.4. Error Control Analysis

**Type II Error.** Figure 6 shows the FNR-$\alpha$ curve. In this paper, we minimize the Type II error while enforcing the upper bound of Type I error at $\alpha$. The performance of Type II error cannot be fully controlled, which mainly depends on how well the score function can quantify model's ability to answer correctly.

Figure 7 illustrates the error control analysis on HotpotQA, highlighting FACTTEST's capability to control the Type I error effectively. Figure 8 presents the Type I error calibration results of FACTTEST-t across four datasets, complementing the experiments discussed in Section 5.3.

## D.5. Additional Results with More Baselines

We include SAPLMA as an additional baseline and further exploit its classifier logit as a classifier-based scoring function, which we denote as FTEST-saplma. For FTEST-saplma, we train the classifier on half of the training set and use the remaining samples for calibration; for the SAPLMA baseline, we train on the full dataset. Table 8 compares FACTTEST 's performance against all the baselines. Note that we omit SAPLMA's accuracy on FEVER with LLaMA-13B, as it classifies all predictions as incorrect in that setting.

## D.6. Additional Results with More Base Models

To further verify the effectiveness of FACTTEST, we additionally evaluate the performance of FACTTEST on more base models, including more pretrained models, instruction-tuned models and black-box models.

Table 9: The accuracy performance (%) of FACTTEST on four question-answering datasets using **Mistral-7B** as the base model. The significance level for FACTTEST is set to 0.1. The percentages inside the parentheses are the Type I error.

| Dataset | Base | SelfCheckGPT | FACTTEST-ve$_{15}$ | FACTTEST-se$_{15}$ | FACTTEST-kle$_{15}$ |
|---------|------|--------------|--------------------|--------------------|---------------------|
| ParaRel | 39.79 | 57.01 (0.25) | 65.63 (0.07) | 70.20 (0.08) | 72.78 (0.08) |
| HotpotQA | 36.48 | 46.01 (0.46) | 61.81 (0.06) | 63.06 (0.05) | 65.59 (0.05) |
| FEVER | 35.47 | 41.76 (0.05) | 22.99 (0.08) | 51.05 (0.08) | - |
| WiCE | 55.85 | 56.24 (0.47) | 68.81 (0.08) | 68.64 (0.08) | - |

- **Pretrained Model.** In Table 9, we employ Mistral-7B (Jiang et al., 2023) as the pretrained base model to supplement the results in Table 1.
- **Instruction-tuned Model.** In Table 10, we employ LLaMA-3.2-3B-Instruct (Dubey et al., 2024) and Tulu2-7B (Ivison et al., 2023) as base models to evaluate the performance of FACTTEST on instruction-tuned models.
- **Black-box Model.** In Table 11, we employ Claude-3.5-Sonnet (Anthropic, 2024) and GPT-4o (OpenAI, 2024b) as base models to evaluate the performance of FACTTEST, with OpenLLaMA-3B serving as the open-source model to calculate

Table 10: The accuracy performance (%) of FACTTEST using Llama-3.2-3B-Instruct and Tulu-2-7B as base models. The $\alpha$ is set to 0.10. The percentages inside the parentheses are the Type I error.

| Dataset | Model | Base | FACTTEST-se$_{15}$ | FACTTEST-kle$_{15}$ |
|---|---|---|---|---|
| ParaRel | Llama-3.2-3B-Instruct | 39.34 | 72.79 (0.08) | 80.01 (0.08) |
| ParaRel | Tulu-2-7B | 43.89 | 75.47 (0.06) | 78.49 (0.07) |
| HotpotQA | Llama-3.2-3B-Instruct | 33.40 | 57.75 (0.06) | 60.38 (0.07) |
| HotpotQA | Tulu-2-7B | 32.91 | 53.54 (0.05) | 45.89 (0.10) |
| WiCE | Llama-3.2-3B-Instruct | 55.11 | 75.16 (0.09) | - |
| WiCE | Tulu-2-7B | 57.20 | 63.22 (0.08) | - |
| FEVER | Llama-3.2-3B-Instruct | 33.33 | 68.48 (0.10) | - |
| FEVER | Tulu-2-7B | 47.87 | 69.40 (0.09) | - |

Table 11: The accuracy performance (%) of FACTTEST on ParaRel using OpenLlama-7B as an open-source model. The significance level is set to 0.1. The percentages inside the parentheses are the Type I error.

| Model | Base | SelfCheckGPT | FACTTEST-se$_{15}$ | FACTTEST-kle$_{15}$ |
|---|---|---|---|---|
| Claude-3.5-Sonnet | 58.25 | 58.96 (0.92) | 73.29 (0.08) | 79.86 (0.08) |
| Gemini-1.5-Flash-8B | 64.23 | 65.92 (0.86) | 76.87 (0.07) | 80.01 (0.08) |
| GPT-4o | 66.39 | 69.71 (0.83) | 80.70 (0.07) | 82.76 (0.08) |

scores.

Our findings reveal that incorporating FACTTEST significantly reduces hallucinations, achieving an average accuracy improvement of 24% while maintaining Type I errors below the specified $\alpha$.

### D.7. Additional Results with More Score Functions

Table 12: The accuracy and answer rate performance (%) of FACTTEST with a significance level $\alpha = 0.1$ using 5-generation and 10-generation KLE as score functions.

| Dataset | Model | Base | FACTTEST-kle$_5$ | FACTTEST-kle$_{10}$ |
|---|---|---|---|---|
| ParaRel | OpenLLAMA-3B | 36.66 | 71.65 (18%) | 74.72 (20%) |
| ParaRel | OpenLLAMA-7B | 40.38 | 72.99 (20%) | 75.90 (20%) |
| Hotpot | OpenLLAMA-3B | 25.72 | 52.34 (11%) | 51.82 (12%) |
| Hotpot | OpenLLAMA-7B | 28.63 | 52.45 (11%) | 55.92 (13%) |

In this section, we present results using additional score functions, including FACTTEST-kle5 and FACTTEST-kle10, to complement the findings in Table 1.

Besides, we implement FACTTEST-scgpt to illustrate how our framework can integrate various uncertainty or hallucination quantification methods. For FACTTEST-scgpt, we utilize the negative value of the SelfCheckGPT-NLI score as the score function. The accuracy and Type I error performance for this implementation are summarized in Table 14.

Moreover, to highlight that our framework extends beyond uncertainty-based approaches, we develop a classifier-based variant, FACTTEST-cls, and compare it with uncertainty-based ones in Table 15. This variant employs a random forest classifier trained on hidden layer activations from both the question and answer, along with probabilistic statistics of the generated answer, to predict the correctness of question-generated answer pairs. The results indicate that FactTest-cls achieves competitive accuracy, maintains Type I error below the specified threshold, and demonstrates improved Type II error rates compared to uncertainty-based score functions.

Table 13: The Type I Error of FactTest with a significance level $\alpha = 0.1$ using 5-generation and 10-generation KLE as score functions.

| Dataset | Model | FACTTEST-kle$_5$ | FACTTEST-kle$_{10}$ |
|---------|-------|------------------|---------------------|
| ParaRel | OpenLLAMA-3B | 0.0783 | 0.0778 |
| ParaRel | OpenLLAMA-7B | 0.0880 | 0.0787 |
| Hotpot | OpenLLAMA-3B | 0.0656 | 0.0643 |
| Hotpot | OpenLLAMA-7B | 0.0643 | 0.0654 |

Table 14: The accuracy and Type I Error performance of FACTTEST-scgpt evaluated on ParaRel with OpenLLaMA-3B serving as the base model.

| $\alpha$ | 0.05 | 0.1 | 0.2 |
|----------|------|-----|-----|
| Accuracy | 61.82 | 62.92 | 59.26 |
| Type I error | 0.04 | 0.09 | 0.17 |

## D.8. Additional Analysis about Unwilling Answered Questions

We perform additional analyses to evaluate the effectiveness of FACTTEST. Table 16 presents the performance of base models on subsets of questions that the model is either unwilling or willing to answer on ParaRel, using FACTTEST-kle15. Notably, the results for the "Willing" subset correspond directly to the performance of FACTTEST-kle15. The results show that accuracy on unwilling samples is significantly lower than on the entire dataset and willing samples, highlighting FACTTEST's capability to decline unknown questions effectively.

## D.9. Score Distribution

Figure 9 represents the certainty distributions of correct subset and incorrect subset using semantic entropy as the score function.

Table 15: The accuracy, Type I Error and Type II Error performance of FACTTEST-cls evaluated on ParaRel with $\alpha = 0.05$.

| Base Model | Metric | FACTTEST-ve15 | FACTTEST-se15 | FACTTEST-cls |
|---|---|---|---|---|
| OpenLLAMA-3B | Accuracy (%) | 67.28 | 67.26 | **85.13** |
| | Type I error | 0.05 | 0.05 | 0.04 |
| | Type II error | 0.86 | 0.85 | **0.35** |
| OpenLLAMA-7B | Accuracy (%) | 80.29 | 65.23 | **89.50** |
| | Type I error | 0.01 | 0.04 | 0.03 |
| | Type II error | 0.92 | 0.87 | **0.44** |
| OpenLLAMA-13B | Accuracy (%) | 79.41 | 73.09 | **88.37** |
| | Type I error | 0.03 | 0.03 | 0.04 |
| | Type II error | 0.91 | 0.87 | **0.42** |

Table 16: The accuracy performance (%) of base models on the subset of questions that the model is unwilling or willing to answer on ParaRel using FACTTEST-kle15. The $\alpha$ is set to 0.1.

| Model | Base | Unwilling | Willing |
|---|---|---|---|
| OpenLlama-3B | 36.66 | 27.90 | 75.51 |
| OpenLlama-7B | 40.38 | 32.93 | 75.36 |
| OpenLlama-13B | 42.21 | 32.81 | 79.55 |

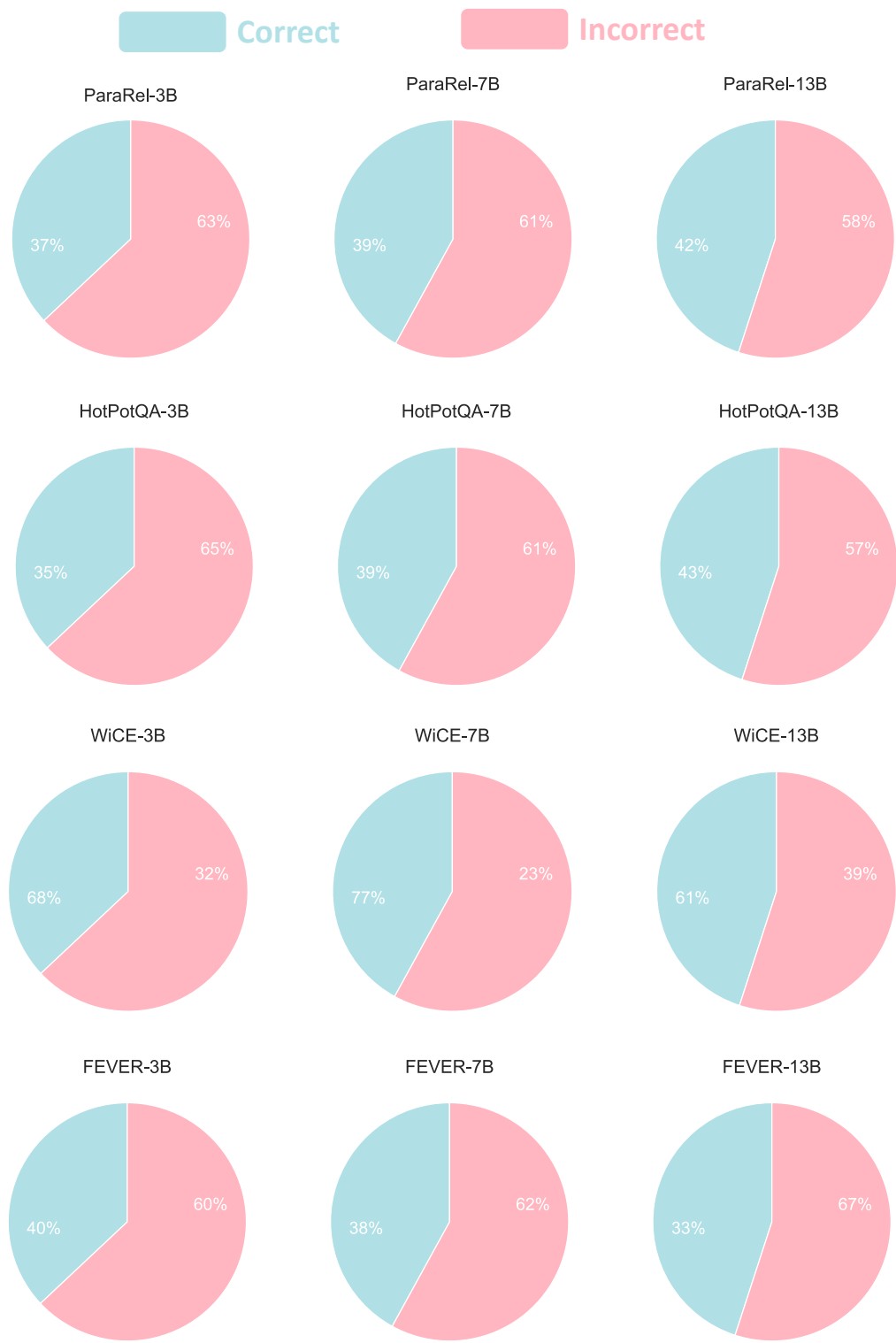

Figure 5: The certain and uncertain data distribution of the originated datasets obtained from supervised identification strategy. The title of each sub-figure consists of the dataset name and the size of the pre-trained model used to evaluate.

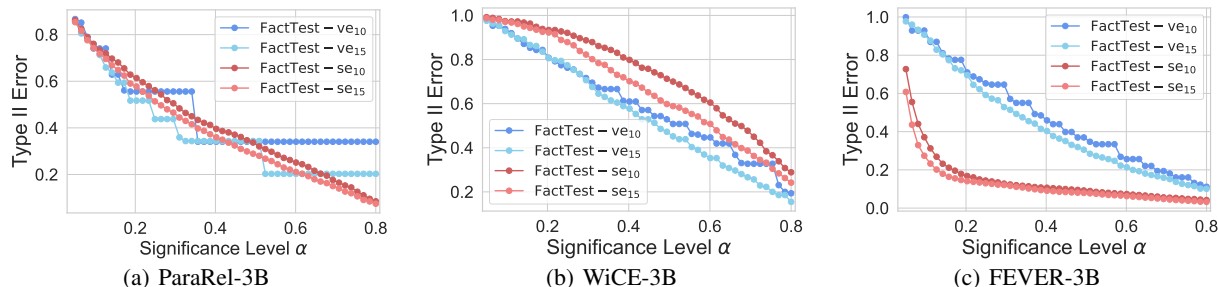

Figure 6: The Type II error, or FNR, of FACTTEST given different significance levels. The caption of each sub-figure consists of the dataset name and the model size.

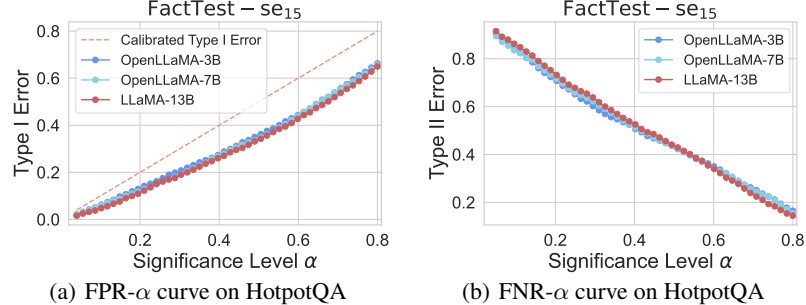

Figure 7: The Type I error and Type II error of FACTTEST given different significance levels on HotpotQA using semantic entropy as the score function. The legend represents the base pretrained model.

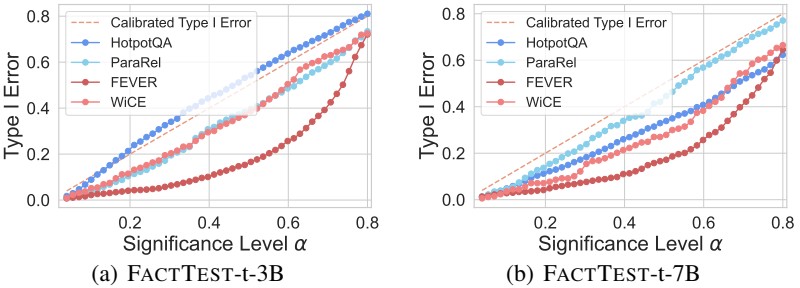

Figure 8: The Type I error calibration results of FACTTEST-t given different significance levels using semantic entropy as the certainty function. The legend represents the dataset name.

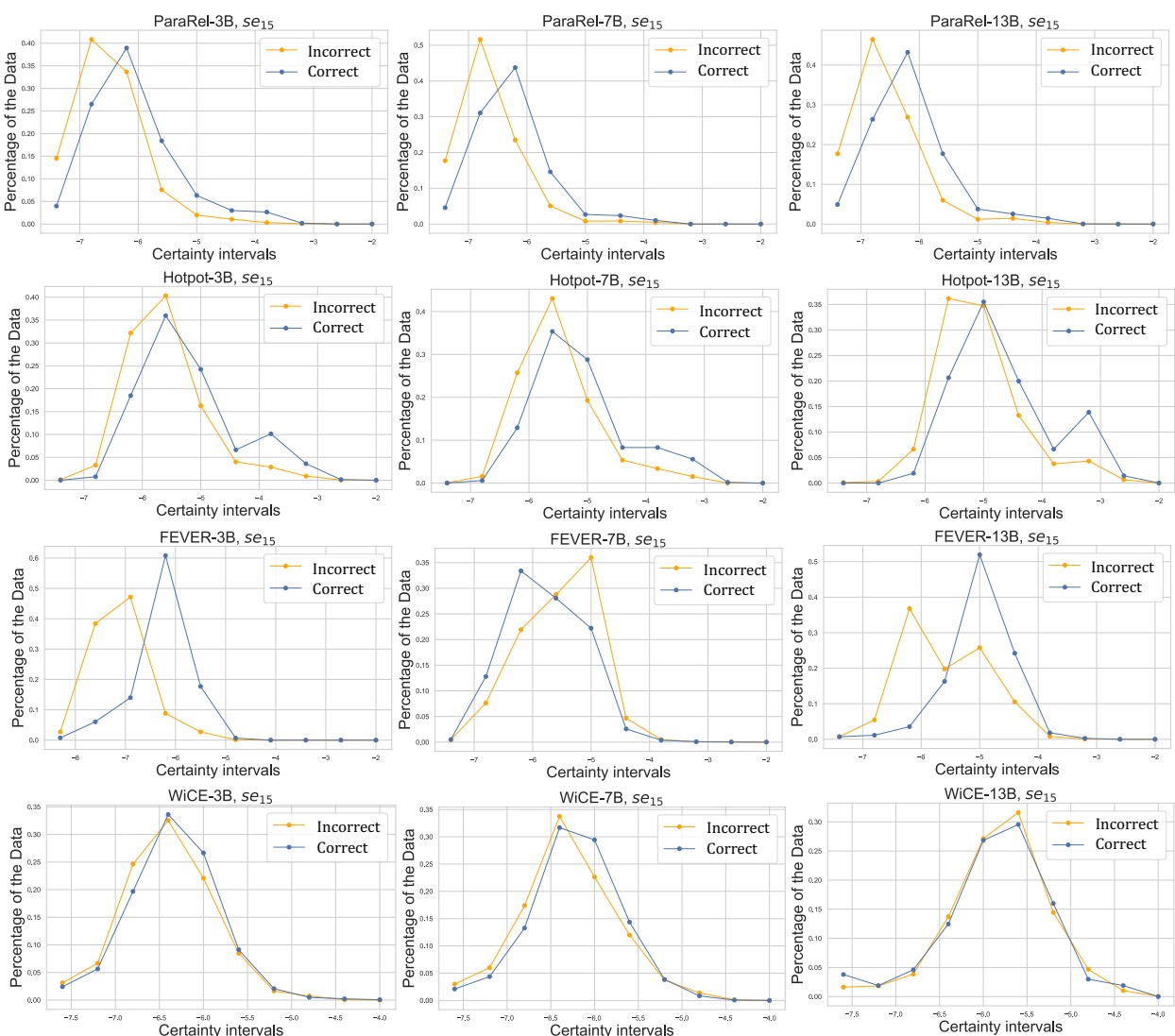

Figure 9: The certainty distribution of the training datasets on certain set and uncertain set. The title of each sub-figure consists of the dataset name, the size of the pre-trained model used to evaluate, the certainty function and the number of generations.

