# OpenReview forum: "FactTest: Factuality Testing in Large Language Models with Finite-Sample and Distribution-Free Guarantees"
_ICML.cc/2025/Conference — ICML 2025 poster_

### Official Review · Reviewer_XhUn · 2025-02-28

**Overall Recommendation:** 4

**Summary:**

• Introduces FACTTEST, a framework that statistically evaluates LLM factuality with theoretical guarantees to detect hallucinations

• Formulates hallucination detection as hypothesis testing, controlling Type I errors (incorrectly classifying hallucinations as truthful) at user-specified significance levels

• Leverages Neyman-Pearson classification techniques to define a score function measuring output correctness and determine appropriate thresholds using calibration datasets

• Provides strong Type II error control (missing actual hallucinations) under mild conditions when the score function effectively captures output correctness

• Extends the framework to handle covariate shifts through density ratio estimation and rejection sampling

• Demonstrates across multiple QA benchmarks that FACTTEST enables models to abstain from answering uncertain questions, improving accuracy by over 40% compared to base models

• Shows FACTTEST outperforms fine-tuned baselines while using less training data

**Claims And Evidence:**

• The Type I error control claim is well-supported by both theoretical analysis (Theorem 2.1) and empirical validation (Table 2, Figure 1) showing error rates consistently below specified significance levels.

• Performance improvement claims are generally substantiated: Table 1 shows significant accuracy gains (e.g., 39.74% to 83.90% on FEVER with OpenLLaMA-3B), though the headline "40% improvement" represents best-case scenarios rather than average gains.

• Comparison with fine-tuned models is supported by Figure 3, demonstrating FACTTEST-t outperforms R-Tuning on HotpotQA and  FEVER. On the other hand, the improvement is significantly smaller on other datasets.

• Black-box API applicability is validated in Table 3, though only tested on GPT-4o mini.

**Essential References Not Discussed:**

N/A

**Experimental Designs Or Analyses:**

• **Type I error control experiments** (Table 2, Figure 1): Thoroughly validated across multiple datasets and models. The calibration process correctly maintains error rates below specified α thresholds, with results properly disaggregated by score function and model size.

• **Comparison with fine-tuned models** (Figure 3): The experimental design fairly acknowledges data usage differences - FACTTEST-t uses only half the training data compared to R-Tuning and Finetune-All, strengthening performance claims.

• **Covariate shift experiments** (Figure 4): Limited to a single dataset (ParaRel-OOD), which is adequate for proof-of-concept but insufficient to fully validate generalizability across different distribution shift types.

• **Black-box API validation** (Tables 3): Sound approach of using open-source models to calculate certainty scores for closed models, though limited testing on question-answering tasks only.

• **Limitations**:
  - No statistical significance testing for accuracy improvements
  - Temperature settings could affect uncertainty estimation
  - Limited analysis of score function selection impact on overall performance
  - No explicit runtime analysis for practical deployment considerations

**Methods And Evaluation Criteria:**

• The evaluation datasets (ParaRel, HotpotQA, WiCE, FEVER) represent a reasonable mix of question-answering and multiple-choice formats, spanning different knowledge domains.

• The metrics (accuracy, Type I error, Type II error) directly align with the paper's goals and theoretical framework, providing clear performance indicators.

• Comparing against both non-training methods (base models, SelfCheckGPT) and training-based approaches (R-Tuning) offers comprehensive benchmarking.

• Testing across model scales (3B to 13B parameters) and architectures helps demonstrate generalizability.

• The evaluation on black-box APIs is particularly valuable for real-world applicability where model internals may be inaccessible.

**Other Comments Or Suggestions:**

N/A

**Other Strengths And Weaknesses:**

**Strengths:**

• Works with any uncertainty quantification method as the score function, making it adaptable as better estimation techniques emerge.

• Can be applied without fine-tuning, providing immediate benefits to existing models.

• Could be incorporated into existing LLM systems.

**Weaknesses:**

• **Limited to classification/short-form settings**: All experiments focus on question-answering or multiple-choice tasks with short responses. No evaluation on long-form generation where hallucinations often manifest differently and may require different detection approaches.

• Multiple generations required for uncertainty estimation could limit practical deployment in latency-sensitive applications.

• Performance likely sensitive to threshold choices, but limited analysis of this sensitivity.

• Potential for false negatives when correct answers differ syntactically from reference answers.

• The framework's binary approach may oversimplify factuality, which often exists on a spectrum.

• Qualitative analysis of when/why the method fails would strengthen understanding of its limitations.

**Questions For Authors:**

1. The framework was evaluated only on QA and multiple-choice tasks. Have you investigated its applicability to long-form generation where hallucinations often manifest differently (e.g., factual inconsistencies within paragraphs)? For example, testing on the dataset from "Long-form factuality in large language models".

2. What is the runtime overhead of FACTTEST compared to base models? The requirement for multiple generations (5-15) for uncertainty estimation could create latency issues in practical applications.

3. In many real-world scenarios, comprehensive calibration datasets with known ground truth may not be available. How might FACTTEST be adapted for open-domain settings where correct answers for calibration are limited?

**Relation To Broader Scientific Literature:**

• **Hallucination detection methods**: FACTTEST bridges the gap between three existing approaches:
  - Retrieval-based methods that require external knowledge bases
  - Training-based approaches like R-Tuning that need extensive fine-tuning
  - Uncertainty estimation techniques that lack theoretical guarantees

• **Statistical learning theory**: Builds directly on Neyman-Pearson classification work

• **Selective prediction**: Advances the line of work on LLM "know when they don't know" capabilities by providing formal statistical guarantees.

• **LLM factuality benchmarks**: Uses established datasets (ParaRel, HotpotQA, FEVER, WiCE) that have been employed in previous factuality research.

**Theoretical Claims:**

I did not look too closely at proofs. A cursory glance indicates they seem correct; the primary limitation is that Type II error control depends on score function quality, which is appropriately acknowledged in the paper.

---

> ### Author Rebuttal · Authors · 2025-03-31
>
> Thank you for your detailed reviews and questions you raise to help improve our paper!
>
> > ***W1: statistical significance testing***
>
> We've conducted bootstrap analysis for 95% confidence intervals. Results available in: https://anonymous.4open.science/r/ICML_rebuttal-8905/icml2025__FactTest-2.pdf. Confidence intervals for all datasets will be included in our revision.
>
> > ***W2: Temperature could affect UQ***
>
> Temperature affects uncertainty quantification but **doesn't limit FactTest**, which controls type I error for any score regardless of temperature. Experiments with different temperatures (see link above) show that while accuracy varies, type I error remains below α.
>
> > ***W3: Score function selection impact***
>
> For any score function $\hat\eta$, the type I error is always controlled below $\alpha$ (Theorem 2.1). Type II error is nearly optimal with an inflation depending on $\epsilon_\eta=\inf_{H\text{ increasing}}\|H\circ\hat\eta-\eta\|_\infty$, which is the deviation between the oracle score $\eta$ and the used score $\hat\eta$ up to increasing transformation $H$ (Theorem 2.2).
>
> > ***W4: Multiple generations and latency***
>
> FactTest works with any score function, including single-generation ones. FactTest-cls in Table 14 (Section D.6) directly predicts answer correctness without multiple generations, with negligible runtime overhead. We expect more efficient score functions developed in the future to further enhance FactTest.
>
> > ***W5: sensitive to threshold choices***
>
> I am afraid there is a misunderstanding of our type I and II error guarantee. Our method selects threshold that guarantees type I error control below α with probability ≥1-δ for any user-specified α,δ. Moreover, our power analysis indicates that the threshold selected always possesses nearly optimal type II error as long as the score function captures the correctness of generated answers. Therefore, **the performance of the selected threshold is always guaranteed**.
>
> > ***W6: False negatives***
>
> FNs occur when score functions poorly separate correct/incorrect samples. This requires better score functions, **orthogonal to our contribution of providing statistical guarantees for any score function**.
>
> > ***W7:binary approach may oversimplify factuality***
>
> We currently use binary accept/reject decisions at significance level α. A natural extension is to output the answer together with the largest confidence level $1-\alpha$ for which the answer is rejected, providing a spectrum of factuality..
>
> > ***W8: when/why fails***
>
> While FactTest controls type I error for any score function, poor score functions increase type II error. With a constant function, FactTest would reject all answers to maintain type I error bounds.
>
> > ***Q1: long-form***
>
> Extending to long-form is our future direction. We see two approaches:
> 1.Document-level analysis: We could apply our framework with score functions designed to measure overall factuality, treating the entire response as a single unit.
> 2.Claim-level analysis: Formulate as a multiple testing problem and extends FactTest from controlling false positive rate to false discovery rate. For an answer with $m$ claims $c_1,...,c_m$, we will have:
> $H_{0,j}$: Claim $c_j$ is not correct.
> $H_{1,j}$: Claim $c_j$ is correct.
>
> > ***Q2: runtime overhead***
>
> Inference Runtime of FactTest shown in https://anonymous.4open.science/r/ICML_rebuttal-8905/icml2025__FactTest-2.pdf
> Single-generation function (e.g., FactTest-cls) add negligible overhead.
>
> > ***Q3: Open-domain settings with limited calibration data***
>
> If the correct answers for in-distribution questions $P_{q,M(q),a}$ are limited, one possiblility is to incoporate OOD questions $\tilde P_{q,M(q),a}$ for which correct answers are available, and then apply our method in section 3 to address the distribution shift in the calibration samples. Our method in section 3 only requires the oracle rule (optimal score function), of judging whether an answer is correct or not for a question, remains the same, $P(y=1|q,M(q))=\tilde P(y=1|q,M(q))$. Then it remains to estimate the density ratio of incorrect answers $\frac{dP_{q,M(q)|y=0}}{d\tilde P_{q,M(q)|y=0}}$, which equals to
> $$\begin{align}
> \frac{dP_{q,M(q)|y=0}}{d\tilde P_{q,M(q)|y=0}}=\frac{dP_{q,M(q),y=0}\tilde P_{y}(0)}{d\tilde P_{q,M(q),y=0}P_y(0)}=\frac{dP_{q,M(q)}\tilde P_y(0)}{d\tilde P_{q,M(q)}P_y(0)}\propto\frac{dP_{q,M(q)}}{d\tilde P_{q,M(q)}}.
> \end{align}$$
> Since the multiplicative constant $\frac{\tilde P_y(0)}{P_y(0)}$ in the density ratio only affect the efficiency of rejection sampling, provided the range $B$ of uniform random variables is large enough compared to $\frac{\tilde P_y(0)}{P_y(0)}$, we can estimate the density ratio $\frac{dP_{q,M(q)}}{d\tilde P_{q,M(q)}}$ based on merely unlabeled question-generated answer pairs $(q,M(q))$, which doesn't rely on the correct answers at all. Therefore, we believe our method is still applicable even if correct answers for in-distribution questions are limited.

---

### Official Review · Reviewer_GerV · 2025-03-13

**Overall Recommendation:** 3

**Summary:**

The paper proposes a framework to provide a statistical guarantee of the correctness of an output generated by LLM. The methodology leverages hypothesis testing and provides guarantees about type I and type II errors. Experiments are conducted on question-answering datasets.

## Update after rebuttal: I have increased my score after authors addressed my concerns.

**Claims And Evidence:**

No issues identified

**Essential References Not Discussed:**

Please refer to mentioned baselines in the strength and weakness section.

**Experimental Designs Or Analyses:**

No issues identified

**Methods And Evaluation Criteria:**

No issues identified

**Other Comments Or Suggestions:**

Please refer to previous section

**Other Strengths And Weaknesses:**

Strengths

- The work focuses on providing statistical guarantees while checking the factuality of the generated content.

- The proposed method performs well for covariate shifts as well.

Weaknesses

- The guarantees are dependent on training data pairs. The generalizability of the proposed method could be more convincing with experiments on the OOD dataset.

- Baselines are limited. To make the experiments more comprehensive, the baselines [1,2,3,4,5] could be included.

[1] Chen, Chao, et al. "INSIDE: LLMs' internal states retain the power of hallucination detection." arXiv preprint arXiv:2402.03744 (2024).

[2] Vashurin, Roman, et al. "Benchmarking uncertainty quantification methods for large language models with lm-polygraph." arXiv preprint arXiv:2406.15627 (2024).

[3] Lin, Zhen, Shubhendu Trivedi, and Jimeng Sun. "Generating with confidence: Uncertainty quantification for black-box large language models." arXiv preprint arXiv:2305.19187 (2023).

[4] Farquhar, Sebastian, et al. "Detecting hallucinations in large language models using semantic entropy." Nature 630.8017 (2024): 625-630.

[5] Azaria, Amos, and Tom Mitchell. "The internal state of an LLM knows when it's lying." arXiv preprint arXiv:2304.13734 (2023)

**Questions For Authors:**

Please refer to previous section

**Relation To Broader Scientific Literature:**

The work is related to hallucination detection and statistical guarantees for a prediction.

**Theoretical Claims:**

No issues identified

---

> ### Author Rebuttal · Authors · 2025-03-31
>
> Thank you for your feedback and suggestions. We are glad that you acknowledge our work’s motivation, method and theoretical claims. Here we provide responses and additional experimental results to address your concerns.
>
> > ***W1:guarantees are dependent on training data pairs***
>
> In Section 3, **we specifically address the covariate shift setting**, which allows the distribution of testing question-answer pairs to differ from the distribution of calibration pairs, provided that the oracle rule (optimal score function) for judging whether an answer is correct remains consistent.
> We have **already incorporated experiments on OOD datasets** in Section 5.4, where Figure 4 demonstrates that FactTest-O (our extension for out-of-distribution domains) effectively controls Type I error and significantly outperforms baseline methods in terms of accuracy. These results empirically validate our theoretical extension to covariate shifts, showing that our framework maintains its statistical guarantees even when applied to question distributions different from those in the calibration set.
>
> > ***W2:More baselines***
>
> Thank you for suggesting additional related works. We believe there may be a misunderstanding about the nature of our contribution. FactTest is **fundamentally a meta-framework that works with any score function** (including uncertainty quantification methods), deriving a statistically guaranteed threshold to determine whether to reject an answer, rather than proposing a new uncertainty scoring mechanism itself.
>
> Regarding the specific papers mentioned:
>
> [1] proposes a metric to evaluate self-consistency that could **serve as a score function within our FactTest framework rather than as a competing baseline**.
>
> [2] is a **benchmarking paper that does not propose a new UQ or hallucination detection method**, but rather evaluates existing approaches with new metrics.
>
> [3] proposes UQ methods for black-box hallucination detection which could **serve as score functions** for FactTest. We have implemented their UDEG approach as a score function within our framework (FactTest-udeg), with results shown below.
>
> [4] is **already implemented** in FactTest as score function, which we denote as FactTest-se in our experiments.
>
> [5] trains a classifier to predict statement truthfulness. We have implemented this as SAPLMA and compared it with FACTTEST. Additionally, we show how it can be used as a score function within our framework (FactTest-saplma).
>
> Here we provide experiments of FactTest compared with [5]. We also implement [5] and udeg in [3] as a score function within FactTest to further demonstrate how these UQ methods could work within our framework.
>
> | Dataset | Base Model | SAPLMA | FactTest-saplma    | FactTest-kle15 |
> | ------- | ---------- | ------ | --- | -------------- |
> | ParaRel | 36.66  |  67.33 (0.24) | 84.77 (0.04) |  78.45 (0.03)     |
> | HotpotQA  | 25.72 |  25.13 (0.02)   | 31.91 (0.04) | 55.35 (0.03)|
>
>
> | Dataset | Base Model | FactTest-udeg5 | FactTest-udeg10  | FactTest-udeg15 |
> | ------- | ---------- | ------ | --- | -------------- |
> | ParaRel | 36.66  |  44.8 (0.04) | 36.53 (0.04) |  36.71 (0.04)     |
>
> These results further demonstrate how existing uncertainty quantification methods can be integrated into our framework, with FactTest providing statistical guarantees on Type I error control while maintaining or improving accuracy.

---

> > ### Comment · Reviewer_GerV · 2025-04-02
> >
> > Thank you for the clarification about the nature of contribution of the paper, along with additional results.
> >
> > I will raise my score from 2 to 3.

---

> > > ### Author Response · Authors · 2025-04-03
> > >
> > > We appreciate the reviewer for the rebuttal comment and raising the score. If there are any additional concerns or suggestions, please let us know, and we will be happy to make further revisions.

---

### Official Review · Reviewer_yUoS · 2025-03-14

**Overall Recommendation:** 3

**Summary:**

The paper proposes FactTest, a framework to assess if an LLM can be factual with high probability correction guarantees. FactTest treats hallucination detection as a statistical hypothesis-testing problem. By doing this, it rigorously controls the maximum allowed Type I error rate ensuring hallucinations are not incorrectly classified as factual content at user-specified significance levels.. Additionally, under mild conditions, FactTest provides strong control over Type II errors, preventing truthful responses from being mistakenly rejected. The framework is distribution-free, making no assumptions about underlying data distributions or the number of human annotations. Moreover, FactTest is model-agnostic and applies equally well to black-box and white-box language models. It is also robust against covariate shifts, maintaining its effectiveness despite changes in input distributions. Extensive experiments conducted by the authors on question-answering benchmarks demonstrate that FactTest effectively detects hallucinations, enabling LLMs to abstain from answering uncertain questions, resulting in accuracy improvements of over 40%.

**Claims And Evidence:**

A lot of the theoretical claims are beyond my expertise. I have focussed some issues regarding the evaluation in the section below.

**Essential References Not Discussed:**

Mentioned in evaluation.

**Experimental Designs Or Analyses:**

The experimental design seems valid.

**Methods And Evaluation Criteria:**

I find the evaluation to be less convincing. I've listed my concerns below.
- In table 1, using selfcheckgpt-NLI to reduce hallucinations using a threshold of 0.5 severely weakens the baseline. The selfcheckgpt paper does not claim that the NLI classifier is calibrated. A threshold of 0.5 could potentially optimize for increased coverage (instead of reduced risk) hence leading to the values not being comparable.
- If the authors claim FactTest can reduce both type 1 and type 2 errors, why are the only comparisons with the baseline using accuracy? Why not use a metric like AUC-PR [1,2], AUCROC[2], or AU risk coverage curve[3] like several other papers that measure uncertainty estimates for factuality? These metrics measure both type 1 and type 2 errors.
- The other tables that measure type 1 and type 2 don't compare to any baselines, so it's hard to know if Facttest is actually more reliable at measuring hallucinations than other approaches.

[1] Manakul, Potsawee, Adian Liusie, and Mark JF Gales. "Selfcheckgpt: Zero-resource black-box hallucination detection for generative large language models." arXiv preprint arXiv:2303.08896 (2023).

[2]Fadeeva, Ekaterina, et al. "Fact-checking the output of large language models via token-level uncertainty quantification." arXiv preprint arXiv:2403.04696 (2024).

[3] Kamath, Amita, Robin Jia, and Percy Liang. "Selective question answering under domain shift." arXiv preprint arXiv:2006.09462 (2020).

**Other Comments Or Suggestions:**

-

**Other Strengths And Weaknesses:**

As mentioned, I've only commented on my concerns regarding the empirical evaluations (the improper metrics, and lack of proper comparisons against baselines).
I am happy to engage in more discussion with the authors about the concerns.

**Questions For Authors:**

Questions listed as concerns regarding evaluation.

**Relation To Broader Scientific Literature:**

The authors have missed more recent papers on estimating uncertainty for factuality.

**Theoretical Claims:**

At a very surface level. Many of the theoretical claims made about the connection between NP classification and PAC-style conformal prediction are beyond my expertise.

---

> ### Author Rebuttal · Authors · 2025-03-31
>
> Thank you for your constructive feedbacks. We are glad that you acknowledge the validity of our framework and experimental design. Here we provide responses to your concerns one by one.
>
> > ***W1:Threshold of SCGPT***
>
> We acknowledge that using a threshold of 0.5 for SelfCheckGPT may not be optimal. In our framework, SelfCheckGPT is more suitable as a score function where thresholds are derived with explicit control over Type I error, rather than as a direct baseline. FactTest aims to determine thresholds that guarantee control over Type I error for any score function. In fact, Table 13 in Section D.6 demonstrates the integration of SelfCheckGPT as a score function in our framework.
> For a more appropriate comparison, we have implemented SAPLMA [1] as a baseline, a classifier specifically designed for hallucination detection that predicts whether an answer is correct. The accuracy (\%) and Type I error (numbers in parentheses) performance are:
>
> | Dataset  | Base Model | SAPLMA       | FactTest-kle15 |
> | -------- | ---------- | ------------ | -------------- |
> | ParaRel  | 36.66      | 67.33 (0.24) | 78.45 (0.03)   |
> | HotpotQA | 25.72      | 25.13 (0.02) | 55.35 (0.03)   |
>
> [1] Azaria, Amos, and Tom Mitchell. "The internal state of an LLM knows when it's lying."
>
> > ***W2:why accuracy, why not AUC-PR, AUCROC, or AURCC***
>
> The primary aim of FactTest is rigorous control over Type I error by explicitly setting a rejection threshold for any score function (including UQ) with statistical guarantees. When determining whether an answer is correct using a given score function, a threshold must be selected to distinguish correct from incorrect responses, with each threshold corresponding to a specific Type I error rate. Rather than reducing Type I error, FactTest determines the threshold for any score function that can statistically ensure the Type I error below user-specified $\alpha$. This operating point is crucial for high-stakes applications where accepting a hallucinated answer even rarely can be unacceptable.
> - Why not AUC-PR, AUCROC, or AURCC: These metrics evaluate the overall effectiveness of uncertainty scores across all possible thresholds. They do not capture performance at the fixed operating point (a specific threshold at a user-specified $\alpha$) required by our method. These metrics would be more appropriate for evaluating the underlying score functions rather than the thresholding mechanism itself. Users could use such metrics to compare different score functions before applying our FactTest framework.
> - Why accuracy: Since we determine the threshold at a user-specified α and use it to reject answers considered incorrect, we naturally report Type I error (to verify our theoretical guarantees) and accuracy on willingly answered questions (to demonstrate practical utility). This approach directly measures the performance at our chosen operating point rather than averaging across all possible thresholds.
>
>
> > ***W3:type 1 and type 2 don't compare to any baselines***
>
> Our framework aims to control Type I error under a user-specified level α, meaning the threshold determined by FactTest depends on α. UQ methods typically output scores without rejection thresholds, while baselines like R-Tuning use fixed rejection rules, resulting in only one Type I error rate.
> Our Type I error figures (Figure 1) demonstrate performance with different α values, showing that Type I error can almost always be controlled below α using FactTest. Adding a baseline like R-Tuning would simply show a horizontal line, as it doesn't offer the flexibility of varying threshold levels that FactTest provides.
>
> > ***W4:more recent papers on uncertainty***
>
> We appreciate this feedback and will incorporate discussion of more recent works in our next version to ensure comprehensive coverage of related methods, including but not limited to:
>
> "[1] proposes a computationally efficient method that leverages semantic diversity to detect hallucinations. [2] revisits standard uncertainty metrics and highlights the limitations of naive entropy-based methods. [3] provides an effective and computationally efficient method to quantify predictive uncertainty in LLMs."
>
> [1] Semantic Entropy Probes: Robust and Cheap Hallucination Detection in LLMs
>
> [2] Rethinking Uncertainty Estimation in Natural Language Generation
>
> [3] Improving Uncertainty Estimation through Semantically Diverse Language Generation

---

> > ### Comment · Reviewer_yUoS · 2025-04-03
> >
> > Thanks for the clarifications!
> > I still recommend weak acceptance

---

### Decision · Program_Chairs · 2025-05-01

**Decision:**

Accept (poster)

**Comment:**

This paper focuses on the fundamental problem of hallucination detection and formulates the problem as hypothesis testing. The proposed method is model-free. The central claim of Type I appears to be well supported and the reviewers appreciated both the importance of the problem as well as the proposed contribution. Therefore, I recommend the acceptance.